# Major sulfonate transporter Soa1 in *Saccharomyces cerevisiae* and considerable substrate diversity in its fungal family

Sylvester Holt[1,2], Harish Kankipati[1,2], Stijn De Graeve[1,2], Griet Van Zeebroeck[1,2], Maria R. Foulquié-Moreno[1,2], Stinus Lindgreen[3,†] & Johan M. Thevelein[1,2]

Sulfate is a well-established sulfur source for fungi; however, in soils sulfonates and sulfate esters, especially choline sulfate, are often much more prominent. Here we show that *Saccharomyces cerevisiae YIL166C(SOA1)* encodes an inorganic sulfur (sulfate, sulfite and thiosulfate) transporter that also catalyses sulfonate and choline sulfate uptake. Phylogenetic analysis of fungal *SOA1* orthologues and expression of 20 members in the *sul1Δ sul2Δ soa1Δ* strain, which is deficient in inorganic and organic sulfur compound uptake, reveals that these transporters have diverse substrate preferences for sulfur compounds. We further show that *SOA2*, a *S. cerevisiae SOA1* paralogue found in *S. uvarum*, *S. eubayanus* and *S. arboricola* is likely to be an evolutionary remnant of the uncharacterized open reading frames *YOL163W* and *YOL162W*. Our work highlights the importance of sulfonates and choline sulfate as sulfur sources in the natural environment of *S. cerevisiae* and other fungi by identifying fungal transporters for these compounds.

[1] Laboratory of Molecular Cell Biology, Department of Biology, KU Leuven, Institute of Botany and Microbiology Kasteelpark Arenberg 31, Flanders, B-3001 Leuven-Heverlee, Belgium. [2] Department of Molecular Microbiology, VIB, Kasteelpark Arenberg 31, Flanders, B-3001 Leuven-Heverlee, Belgium. [3] Carlsberg Research Laboratory, Gamle Carlsberg Vej 4, 1799 Copenhagen V, Denmark. † Present address: Steno Diabetes Center A/S, Niels Steensens Vej 2–4, 2820 Gentofte, Denmark. Correspondence and requests for materials should be addressed to J.M.T. (email: johan.thevelein@mmbio.vib-kuleuven.be).

Soil is a common environmental habitat for fungi, in which organic sulfonates and sulfate esters account for the majority of sulfur sources available for microbial growth[1]. Inorganic sulfate, for example, in the form of $SO_4^{2-}$, accounts for only a few per cent of the sulfur available in the soil. In spite of their significance as a natural source of sulfur, sulfonates and sulfate esters have received relatively little attention, especially compared with sulfate. Sulfonates have a sulfonic acid group covalently attached to carbon and can originate from plants, animals and microbial conversion of inorganic sulfate[2–4]. The β-amino sulfonic acid taurine is widely distributed in nature and even makes up 0.1% of the human body weight[5]. Contrary to animals that rely on inorganic sulfur and sulfur-containing amino acids as the source of sulfur, microorganisms can convert sulfonates to sulfite, an intermediate in the sulfate utilization pathway[6,7]. The Saccharomyces cerevisiae α-ketoglutarate dioxygenase Jlp1 can degrade the aliphatic sulfonates, isethionate and taurine, and is the only known fungal gene involved in sulfonate metabolism[8]. Its occurrence in hemiascomycete yeasts correlates with their ability to assimilate aliphatic sulfonates[9]. Sulfate esters are commonly found in soils with, for instance, animal excreta and decaying animal matter as a major origin. Choline sulfate is a major non-acid storage form of sulfate in soils and is highly resistant to nonenzymatic hydrolysis. It is returned to soils from a variety of sources, including plants, fungi and lichens, as well as bacteria, which often secrete it into the medium. It is considered an important natural source of sulfur, as even fungi able to synthesize choline sulfate are also able to take it up from the environment[1].

Up to now, no conclusive experimental evidence has been reported for any fungal sulfonate transporter. In Basidiomycota and Eurotiomycete filamentous fungi, orthologues of the S. cerevisiae JLP1 gene were found to be genomically co-localized with a predicted transporter gene, suggested to encode a sulfonate (taurine) transporter[10]. In the S. cerevisiae genome, JLP1 and YIL166C, an orthologue of this candidate sulfonate transporter gene, are located on different chromosomes. In spite of this, our work has identified the YIL166C gene product as an experimentally confirmed sulfonate transporter in fungi and we renamed the gene, SOA1 (for sulfonate transport). The closest known orthologue of SOA1 is the Aspergillus nidulans alternative sulfate transporter (encoded by astA), isolated in a screen for restoration of growth with sulfate in a mutant strain defective in sulfate transport[11,12]. Its designation as an alternative sulfate transporter refers to the fact that the AstA transporter does not belong to the well-known SulP family of sulfate permeases, to which also the S. cerevisiae Sul1 and Sul2 sulfate transporters belong. Instead, the astA and YIL166C/SOA1 encoded transporters belong to the extensive Dal5 allantoate permease family of fungal major facilitator superfamily (MFS) transporters responsible for uptake of various organic anions and vitamins[12]. In prokaryotes, several experimentally confirmed sulfonate carriers belonging to the ABC class of transporters have been reported[6]. Bacterial transporters are mainly encoded by genes located together with functionally related genes in operons[6] and, based only on this co-localization, putative sulfonate MFS transporters have been suggested for isethionate and sulfoacetate in the soil bacterium Cupriavidus necator[13,14].

In S. Cerevisiae, inorganic sulfur uptake mainly occurs by the high-affinity sulfate $H^+$-symporters, Sul1 and Sul2, which belong to the SulP family and of which the expression is strongly enhanced under sulfur starvation conditions[15–18]. They also transport other sulfate esters and sulfite as well[19,20]. A sul1Δ sul2Δ strain is unable to grow on media with low levels of sulfate as the sole sulfur source, but it can grow in the presence of high levels of sulfate and with sulfite. The responsible carrier(s), however, for growth of this strain with high levels of sulfate and with sulfite have remained elusive. We have recently shown that both Sul1 and Sul2 act as transporter receptors or transceptors for sulfate-induced activation of the protein kinase A (PKA) pathway in sulfur-starved S. cerevisiae cells[20], which prompted us to identify the remaining unknown inorganic sulfur carrier(s).

Here we show that the YIL166C gene product is the remaining inorganic sulfur transporter that is responsible for the uptake of sulfite and high levels of sulfate during growth of the sul1Δ sul2Δ strain with these compounds as the sole sulfur source. Further analysis, however, surprisingly reveals that YIL166C encodes the major sulfonate permease in S. cerevisiae and transports a wide variety of naturally occurring sulfonates at physiological concentrations. Hence, we name YIL166C as SOA1 (for sulfonate transport). Phylogenetic analysis reveals that SOA1 belongs to an extensive family of poorly characterized fungal transporters of organic anions, of which some members were suggested to function as sulfonate transporters. We have expressed multiple SOA1 orthologues from five representative fungal species in S. cerevisiae, which confirms their function in transport of various sulfonates with a diverse specificity. Finally, we show that the S. cerevisiae genes, YOL163W and YOL162W, are likely to be evolutionary leftovers from a second sulfonate transporter gene, which we named SOA2, and which is present in an intact form only in the most distantly related Saccharomyces species, whereas it is completely absent in more closely related Saccharomyces species.

## Results

**YIL166C encodes a sulfur compound transporter.** Deletion of both the SUL1 and SUL2 genes, encoding the high-affinity sulfate transporters in S. cerevisiae, abolishes growth on low levels of sulfate but still permits growth at high concentrations of sulfate (refs 16,20; Fig. 1a). To identify the remaining low-affinity sulfate transporter(s), we have deleted several candidate genes in the sul1Δ sul2Δ strain. They included the paralogues YGR125W and YPR003C, which display only weak sequence similarity to SUL1 and SUL2, and are divergent from the major groups of sulfate transporters[17] and the uncharacterized MFS transporter gene YIL166C (later named SOA1), a gene that is highly upregulated during sulfur starvation[18]. The triple deletion mutants sul1Δ sul2Δ ygr125wΔ and sul1Δ sul2Δ ypr003cΔ displayed similar growth behaviour as the sul1Δ sul2Δ strain with high sulfate (40 mM), sulfite (0.2 mM) or thiosulfate (0.2 mM) as the sole source of sulfur (Fig. 1a). Growth of the sul1Δ sul2Δ strain with sulfite as the sole source of sulfur has previously been reported[21]. On the other hand, the sul1Δ sul2Δ yil166c(soa1)Δ strain lacked significant growth with any of these sources of sulfur, whereas it showed, similar to the other strains, normal growth with methionine (Fig. 1a).

In growth experiments with liquid medium containing sulfite or thiosulfate as the sole source of sulfur, both the sul1Δ sul2Δ and soa1Δ strains were able to grow slowly at concentrations between 1.0 and 7.5 μM, indicating that the two systems were able to take up very low concentrations of these inorganic sulfur compounds (Fig. 1b,c). The growth rate with these compounds was limited by their concentration below 200 μM and showed saturation kinetics, with a $K_s$ (half-velocity constant) of $67.4 \pm 8.3$ and $68.3 \pm 10.3$ μM (s.e.m., $n = 3$) for growth with sulfite of the sul1Δ sul2Δ and the soa1Δ strain, respectively, and a $K_s$ of $75.8 \pm 6.0$ and $11.2 \pm 0.9$ μM (s.e.m., $n = 3$) for growth with thiosulfate of the sul1Δ sul2Δ and the soa1Δ strain, respectively (Fig. 1e,f). In case of the soa1Δ strain, the $K_s$ value is the result of the transporter kinetics of both Sul1 and Sul2. Our results are consistent with both Soa1 and Sul1/Sul2 being high-affinity sulfite

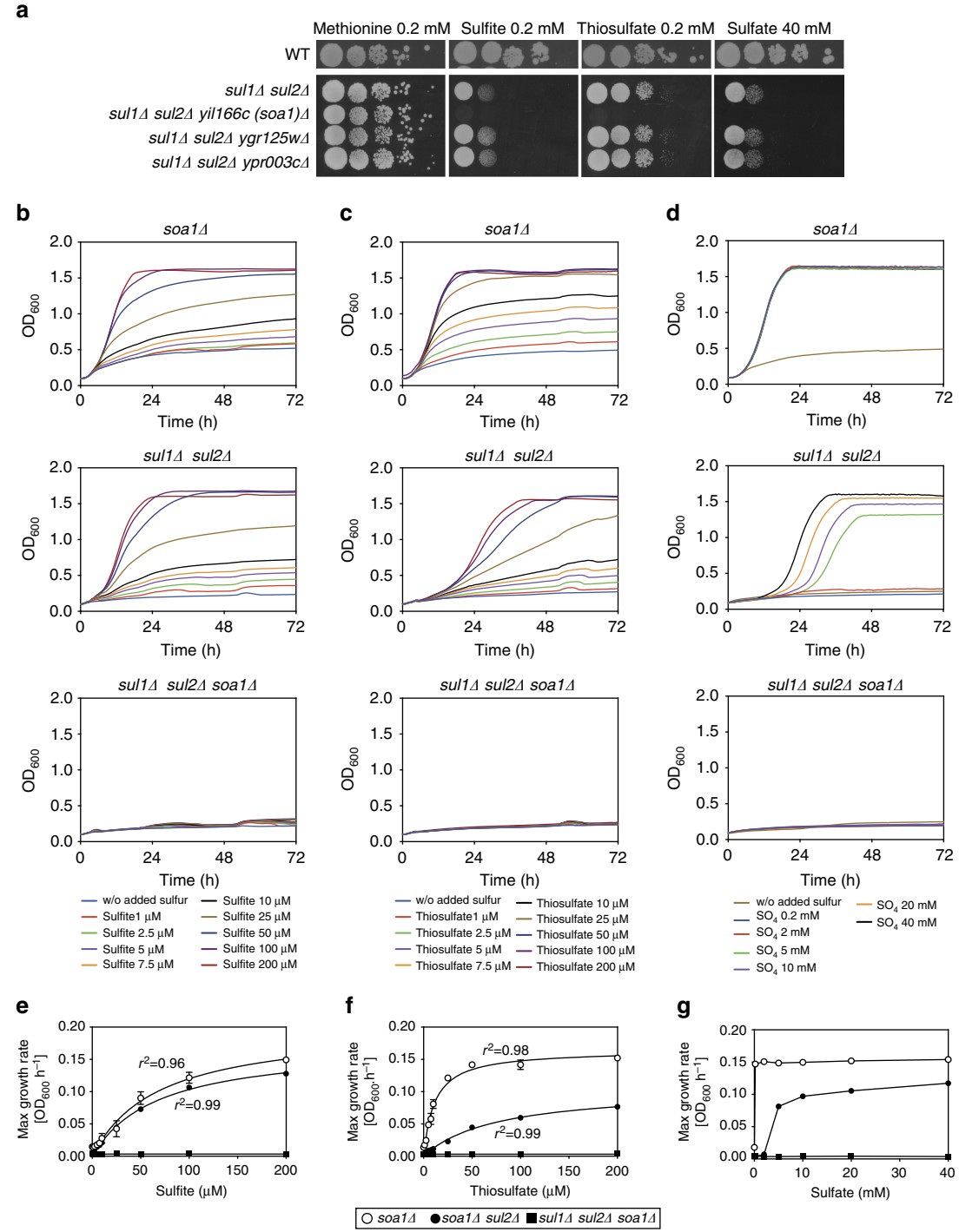

**Figure 1 | Identification of *SOA1* as a sulfur compound transporter gene.** (**a**) *SOA1*, *YGR125W* and *YCR003C* were deleted in a *sul1Δ sul2Δ* strain to create triple deletion strains that were assayed for growth on Sulfur B plates at pH 6.5 with sulfate (40 mM), sulfite (0.2 mM) or thiosulfate (0.2 mM). Growth of the wild-type strain was assayed on different plates because of interference with the growth of the deletion strains by an unidentified sulfur compound (possibly $H_2S$) excreted by the wild-type strain. (**b–d**) The effect of *SUL1*, *SUL2* and/or *SOA1* deletion on growth was assayed in liquid cultures with varying concentrations of (**b**) sulfite (1–200 μM), (**c**) thiosulfate (1–200 μM) and (**d**) sulfate (0.2–40 mM). (**e–g**) Maximal growth rates ($OD_{600}$ $h^{-1}$) were determined from the growth curves and fitted to saturation kinetics (Monod equation) for (**e**) sulfite, (**f**) thiosulfate and (**g**) sulfate. For determination of the half-velocity constants ($K_s$) of sulfite and thiosulfate, maximal growth rates were derived from individual replicates of which the mean is shown with error bars indicating s.d. These experiments were carried out in triplicate from aliquots of the same cell preparation assayed independently. As the experiments with sulfate were carried out in duplicate, error bars are only indicated for sulfite and thiosulfite, which in many cases are so small that they are not visible.

and thiosulfate transporters. In growth experiments with different concentrations of sulfate, from 0.2 to 40 mM, the *sul1Δ sul2Δ soa1Δ* strain was unable to grow at whatever concentration, whereas the *soa1Δ* strain was able to grow equally well at all concentrations and the *sul1Δ sul2Δ* strain grew only from a concentration of 5 mM on and not at all at 2 mM or below

(Fig. 1d,g). These results indicate that *YIL166C(SOA1)* encodes the only the remaining low-affinity sulfate transporter and also the only remaining high-affinity sulfite and thiosulfate transporter in *S. cerevisiae*.

We have characterized growth with sulfite in more detail. Sulfite is present in different forms depending on the pH: $SO_2$, $HSO_3^-$ or $SO_3^{2-}$. At pH 3–5, there is a high level of $SO_2$, which diffuses through the plasma membrane and is converted into $HSO_3^-$ and $SO_3^{2-}$ at the higher intracellular pH[21,22]. Sulfite can be exported by the Ssu1 efflux pump[23–25]. As expected, deletion of *SUL1*, *SUL2* and *YIL166C(SOA1)* did not prevent growth at pH 3.5–4.5, whereas it was strongly reduced at pH 5.5 and absent at pH 6.5 and 7.0 (Fig. 2a). Interestingly, deletion of *SUL1* and *SUL2* had less effect on growth at all pH values compared with deletion of *YIL166C(SOA1)* (Fig. 2a), indicating that *YIL166C(SOA1)* encodes the most active sulfite transporter in *S. cerevisiae*.

**YIL166C encodes a sulfonate transporter**. Given the ability of yeasts to grow on organic sulfur sources and their frequent isolation from soil, we also tested a possible role of *YIL166C(SOA1)* in the uptake of organic sulfur compounds. For that purpose, we tested the *yil166c(soa1)Δ* strain for growth with C2–C7 aliphatic sulfonates and with the substituted sulfonates isethionate, taurine, L-cysteic acid, sulfoacetic acid and hydroxymethane sulfonate as the sole source of sulfur. Deletion of *YIL166C(SOA1)* abolished growth with all these compounds, except for hydroxymethane sulfonate for which there was only a partial reduction in growth (Fig. 2b). The remaining growth with the latter compound was due to the Sul1 and Sul2 transporters, as it was abolished by additional deletion of *SUL1* and *SUL2* (Fig. 2b). We also tested growth with the sulfate ester choline sulfate and, in this case, growth was only abolished in the *sul1Δ sul2Δ yil166c(soa1)Δ* strain, indicating that this compound can be taken up by Sul1, Sul2 and Yil166c(Soa1) (Fig. 2b). As the *YIL166C* encoded transporter seems to play a major role in sulfonate uptake, we called it *SOA1* (for sulfonate transport).

We have also measured the transport kinetics of Soa1 using short-term uptake of radioactively labelled isethionate and taurine in cells that had been starved for sulfur for 3 days, to establish high expression of *SOA1*. Soa1 showed Michaelis–Menten kinetics for uptake of isethionate with medium affinity ($K_m = 116.9 \pm 13.2\,\mu M$; s.e.m., $n = 3$). We have added a Hanes–Woolf plot as inset in Fig. 3a. In the absence of Soa1, isethionate uptake was negligible (Fig. 3a). When isethionate uptake was measured as a function of pH, there was a steep increase at pH values below 6, which was also observed in the *soa1Δ* strain, suggesting that it may be due to diffusion, similar to the uptake of sulfite at low pH. The pH profile of isethionate uptake was virtually identical in the *soa1Δ* and *sul1Δ sul2Δ soa1Δ* strains (Fig. 3a), indicating that neither Sul1 nor Sul2 play any significant role in isethionate uptake at whatever pH of the medium. Isethionate uptake by Soa1 drops from pH 6 to pH 5, which probably reflects a regular pH effect on protein structure/activity, similar to the pH profile of sulfate uptake, which does not suffer from interference with diffusion (see further, Fig. 3d).

For taurine uptake, no sign of saturation was observed up to the maximally tested concentration of 3.6 mM (Fig. 3b). For concentrations below 1.5 mM, taurine uptake was largely dependent on Soa1; at higher concentrations, there was significant residual uptake in the *soa1Δ* strain (Fig. 3b). Taurine is unlikely to diffuse through cell membranes because of its zwitterionic nature[5] and the carrier(s) responsible for the residual taurine uptake remains unknown. Determination of the uptake of a high level of sulfate (40 mM) in the *sul1Δ sul2Δ* and *sul1Δ sul2Δ soa1Δ* strains indicated that the three carriers are responsible for

all the uptake of high sulfate, whereas Sul1 and Sul2 are the major high-capacity carriers for high levels of sulfate, as their deletion strongly curtails the uptake of 40 mM sulfate (Fig. 3c).

**Soa1 and *Aspergillus* AstA are proton symporters**. The closest orthologue of Soa1 that has been experimentally characterized to some extent is the *A. nidulans* alternative sulfate MFS transporter encoded by the *astA* gene[12]. We have expressed the *astA* gene from the *A. nidulans* IAM 2006 strain in *S. cerevisiae*, replacing the *SOA1* gene in the original locus of the *sul1Δ sul2Δ soa1Δ* strain. This was performed by clustered regularly interspaced short palindromic repeat (CRISPR) targeting of the nourseothricin marker (see Methods) and using a codon-optimized synthetic gene as donor DNA. Using the same method, the *S. cerevisiae* SOA1 gene was re-inserted in the *sul1Δ sul2Δ soa1Δ* strain at the original locus, to serve as a control strain. Although the *astA* gene from *A. nidulans* was codon optimized and inserted in the original *SOA1* locus, we cannot exclude that the protein lacks structural modifications happening in the original host or undergoes new modifications in *S. Cerevisiae*, which may alter its kinetics.

We performed sulfate uptake experiments with the *astA*-expressing *S. cerevisiae* strain and the *SOA1* re-insertion strain (Fig. 3d). This showed that AstA is a high-affinity sulfate transporter ($K_m = 7.50 \pm 0.53\,\mu M$; s.e.m., $n = 3$), and that in this low concentration range (up to 200 μM sulfate) Soa1 shows negligible sulfate transport (at whatever pH tested). We have added a Hanes–Woolf plot as inset in Fig. 3d. There is very little information on kinetic characteristics of sulfur compound transporters in fungi. An early paper by Tweedie and Segel[26] reports a $K_m$ value of 75 μM for sulfate uptake in sulfur-limited *Aspergillus* cells; however, it is unclear what transporters were responsible for this uptake. A recent paper by Pilsyk *et al.*[27] has determined the kinetics of long-term sulfate assimilation into cellular organic matter and it is unclear how the kinetic constants determined relate to those of sulfate transport as measured in very short-term uptake experiments.

At a sulfate concentration of 200 μM, AstA displayed a higher pH optimum (pH 7) compared with that of Sul1 and Sul2 (pH 5) (Fig. 3d). High-affinity sulfate transport by AstA displayed a $V_{max}$ of $1.01 \pm 0.02$ nmol min$^{-1}$ per mg cell dry weight (s.e.m., $n = 3$), which was 12.3 times more efficient than sulfonate transport by Soa1, which had a $V_{max}$ of $81.8 \pm 3.6$ pmol min$^{-1}$ per mg cell dry weight (s.e.m., $n = 3$). This comparison assumes similar expression levels at the plasma membrane. This cannot be guaranteed in spite of the fact that the astA protein sequence was codon optimized for expression in *S. cerevisiae* and engineered into the *soa1* locus, ensuring the use of the same promoter and terminator, and at the same chromosomal position. Treatment with carbonyl cyanide 3-chlorophenylhydrazone (CCCP), a compound that abolishes the proton gradient over the plasma membrane, virtually eliminated isethionate uptake by Soa1 and sulfate uptake by AstA, with Soa1 being somewhat less sensitive than AstA (Fig. 3d). This is consistent with the transporters functioning as H$^+$-symporters.

**Bayesian phylogeny of fungal orthologues**. With the new *S. cerevisiae* sulfonate transporter Soa1 and the *A. nidulans* high-affinity alternative sulfate transporter AstA, we have made a phylogenetic tree using all related putative transporters of sulfur compounds in fungi (Fig. 4a, Supplementary Figs 1 and 2, and Supplementary Data 1). Based on the apparent evolutionary gaps and the taxonomic classes, the orthologues could be classified into five distinct groups (Fig. 4a). Group 1, which contains all *astA*-related genes, is clearly distinct from all the other

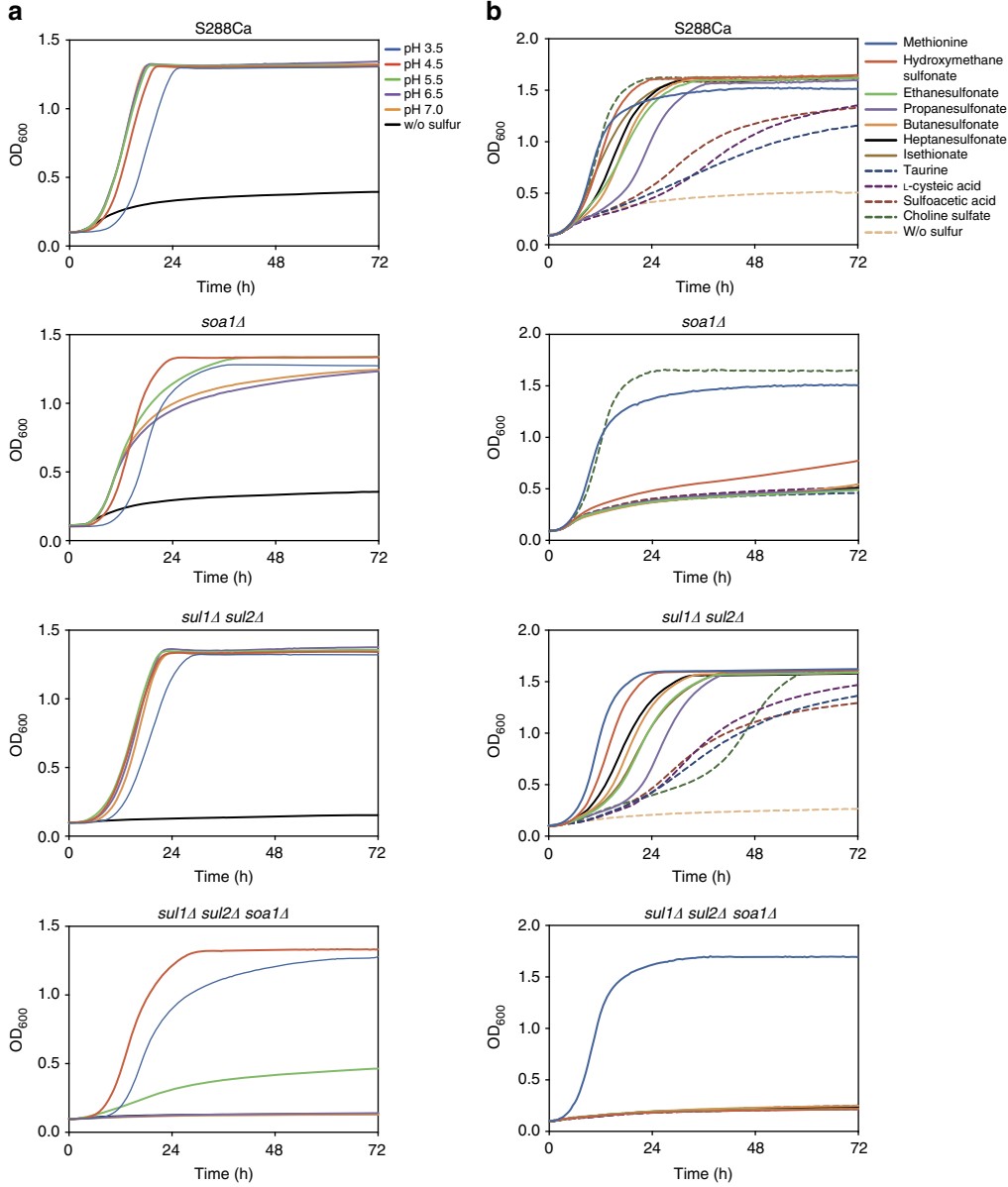

**Figure 2 | pH dependence and substrate diversity.** (**a**) pH dependence of growth with sulfite in malic acid–phosphate-buffered sulfur B medium with the pH adjusted between 3.5 and 7.0. (**b**) Dependence on *SOA1* and/or *SUL1* and *SUL2* for growth with various sulfonates or choline sulfate as sole sulfur source.

*SOA1*-related groups; Group 2 contains the closest orthologues to *S. cerevisiae SOA1* in the other *Saccharomyces sensu stricto* species and in other fungi; Group 3 mainly contains orthologues of Soa2, which is encoded by a paralogous gene of *SOA1* in *Saccharomyces uvarum* and *Saccharomyces Eubayanus*, and of which the gene products show 40% and 41% identity with the gene product of *S. cerevisiae SOA1*, respectively. *SOA2* is also present as an intact gene in the majority of the *Saccharomycetes* yeasts, with many *Candida* species having multiple paralogues; Group 4 contains orthologues from most *Aspergillus* species and in particular orthologues from other *Eurotiomycete* species and from *Dothideomycete* species; Group 5 is the largest group and is especially rich in *Basidiomycota* and *Sordariomycete* orthologues; Group 6 is a very small group that is most distantly related to the other groups, suggesting that it branched off the earliest in evolution. Groups 4 and 5 have been collapsed in Fig. 4a for simplicity and are shown in full in Supplementary Figs 1 and 2. The *SOA1* orthologues were remarkably conserved over all the

fungal taxonomic groups (Fig. 4a and Supplementary Figs 1 and 2), ranging from the Chytridiomycota that originated in the Precambrian time ∼630 million years ago as the earliest fungi to the more recent groups of Basidiomycota and Ascomycota that diverged ∼500 million years ago[28,29].

The 1,011 unique genes were present in 356 strains from 254 species in 147 genera, indicating multiple gene duplication events as previously described[30]. The highest number was present in the *Brassica* plant pathogen *Verticillium longisporum* with 18 paralogous genes. The duplications were highly variable within the genera, for example, in *Aspergillus* and *Fusarium*. In addition to the strain-specific duplications, we also observed mosaic-like patterns in the sense that sometimes orthologues from the same taxonomic group clustered closely together, whereas in other cases orthologues from different taxonomic groups clustered closely together. This was most prominent in Group 1, the *astA*-related group, in which most orthologues were derived from the two *Sodariomycete* genera *Fusarium* and *Verticillium*. On the

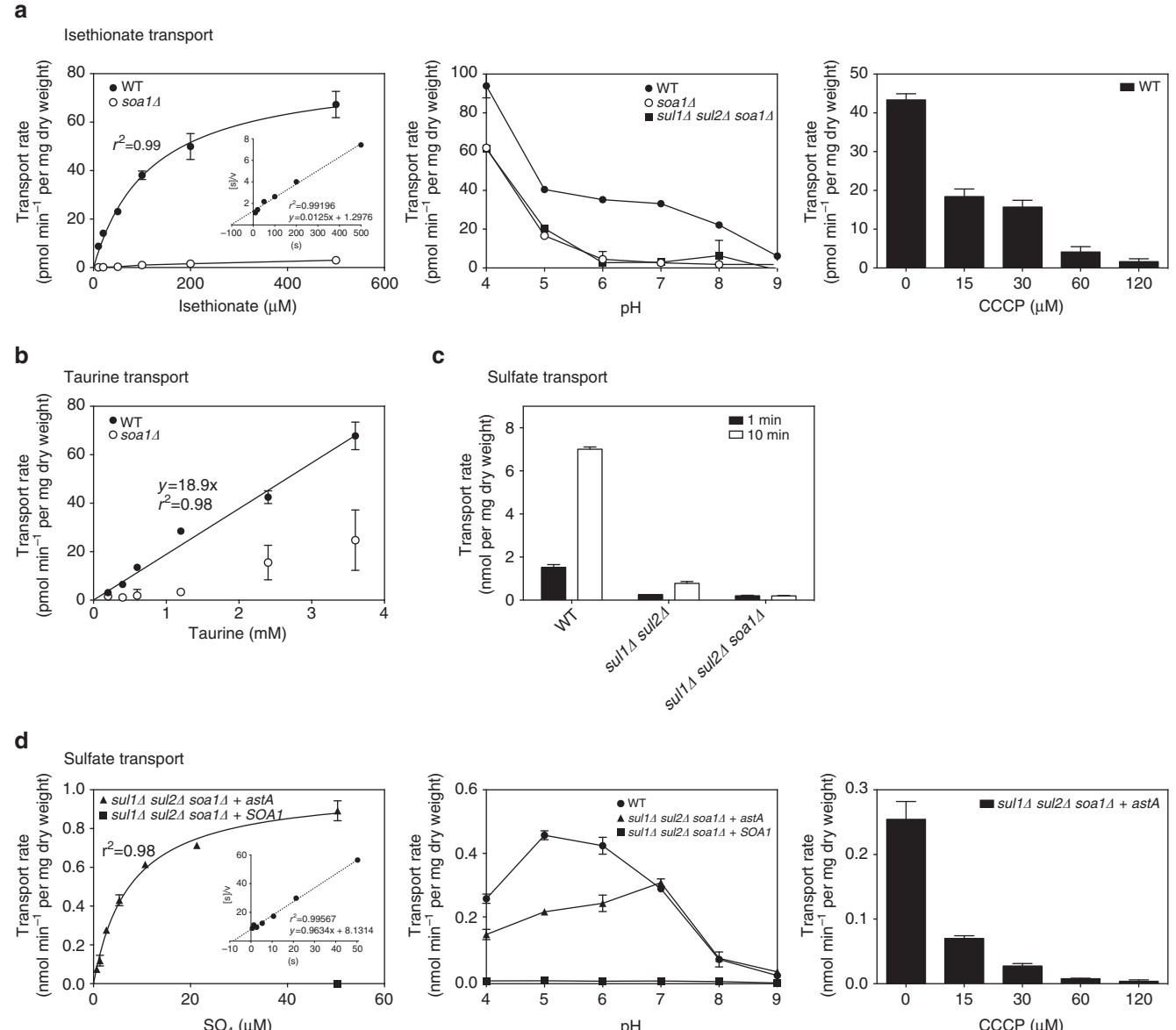

**Figure 3 | Transport kinetics of Soa1 and ASTA.** Cells were starved for sulfur, buffered in Bis-Tris pH 6.5 and transport rates were measured by short-term exposure to different concentrations of radiolabelled isethionate (**a**), taurine (**b**) and sulfate (**c,d**). For low-affinity sulfate transport, the concentration was kept constant at 40 mM and transport was measured for 1 and 10 min (**c**). The affinity constants calculated using the reciprocal approximation of the Hanes–Woolf plots shown as insets ($K_m$ of Soa1: 103.8 μM and $K_m$ of astA: 8.5 μM) are slightly different from the more precise nonlinear fit to the Michalis–Menten equation already reported ($K_m$ of Soa1: 116.9 μM and $K_m$ of astA: 7.5 μM). The pH profiles were assayed in different buffers with adjusted pH, while keeping the amount of isethionate (500 μM) and sulfate (200 μM) constant (A,D). H$^+$-symport activity was tested by exposing cells to the proton gradient decoupling agent CCCP for 10 min before transport measurements (**a,d**). All experiments were carried out in triplicate from aliquots of the same cell preparation assayed independently with the mean shown and error bars indicating s.d. In many cases, the error is so small that the error bars are not visible.

other hand, this group only contained four orthologues from the *Aspergillus* genus, whereas many members of this genus were present in the other groups. This shows that the *astA* gene has been duplicated preferentially in a few genera, while remaining scarce or being functionally lost in other genera, suggesting adaptation to the presence of different sulfur compounds in specific ecological niches.

**Expression of putative sulfonate carriers in yeast.** Next, we tested whether selected orthologues of *SOA1* could complement the absence of sulfur compound uptake in the *sul1Δ sul2Δ soa1Δ* strain, when inserted into the *SOA1* locus. We have selected

orthologues from all major groups and expressed most paralogues from the same organism in *S. cerevisiae*. In addition to *S. cerevisiae SOA1*, we selected *SOA1* and *SOA2* from *S. uvarum* and *S. eubayanus*. The selected genes were inserted at the *SOA1* locus by CRISPR targeting of the nourseothricin marker in the strain *sul1Δ sul2Δ soa1Δ*. The 20 transformants carrying each a different putative sulfur compound transporter, plus the untransformed *sul1Δ sul2Δ soa1Δ* strain, were assayed for growth in liquid medium containing one of 15 sulfur compounds as the sole sulfur source. Figure 4b summarizes the results for the maximum growth rate μ of all strain/sulfur compound combinations, whereas experimental results, lag phases and

maximal densities are available in Supplementary Data 2. The transformants were grown with a selected sulfur source on minimal medium maximally depleted for any sulfur source. The residual sulfate concentration was ~1.4 μM, which was enough to sustain limited residual growth with two orthologues from the

*astA* group, including *astA* from the *A. nidulans* IAM 2006 strain, but not with *SOA1* from *S. cerevisiae* (Supplementary Fig. 3).

The representatives from Group 4 and 5 generally showed poor growth with any of the inorganic and organic sulfur sources (except with the amino acid methionine), whereas in the groups

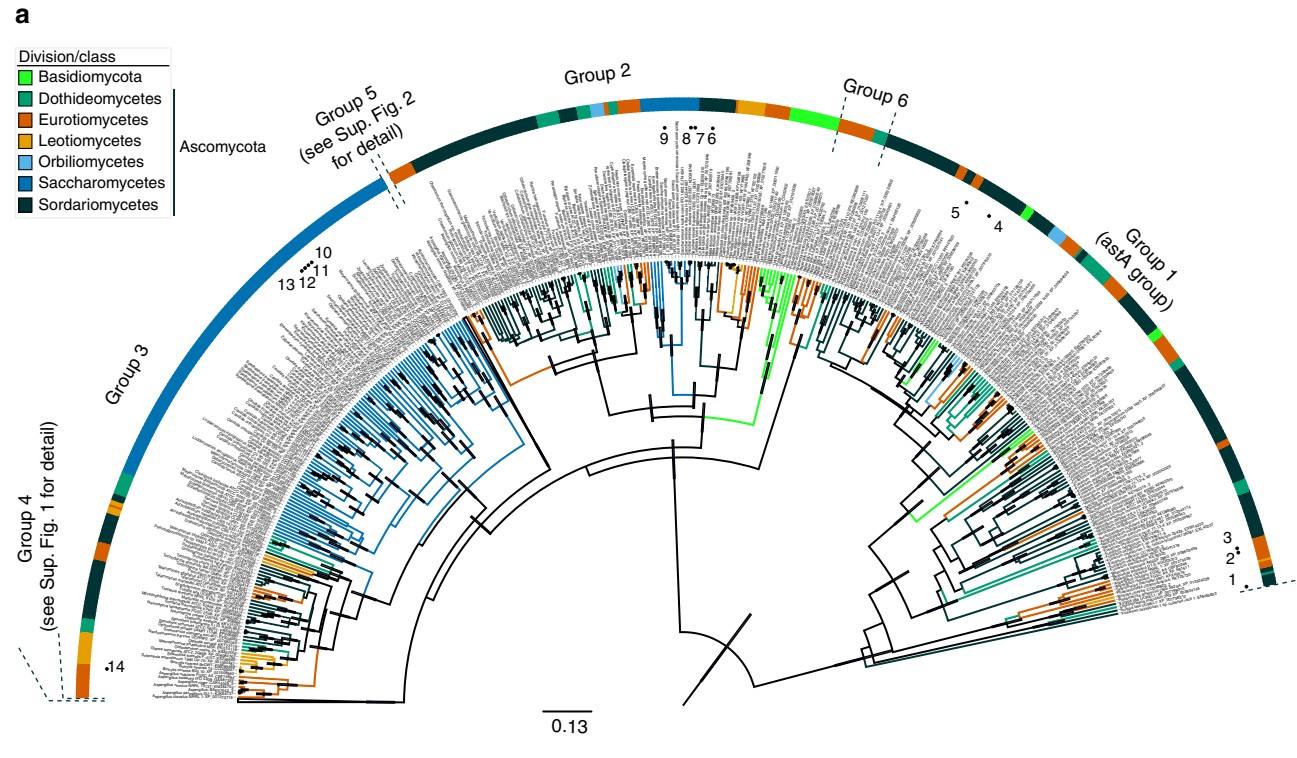

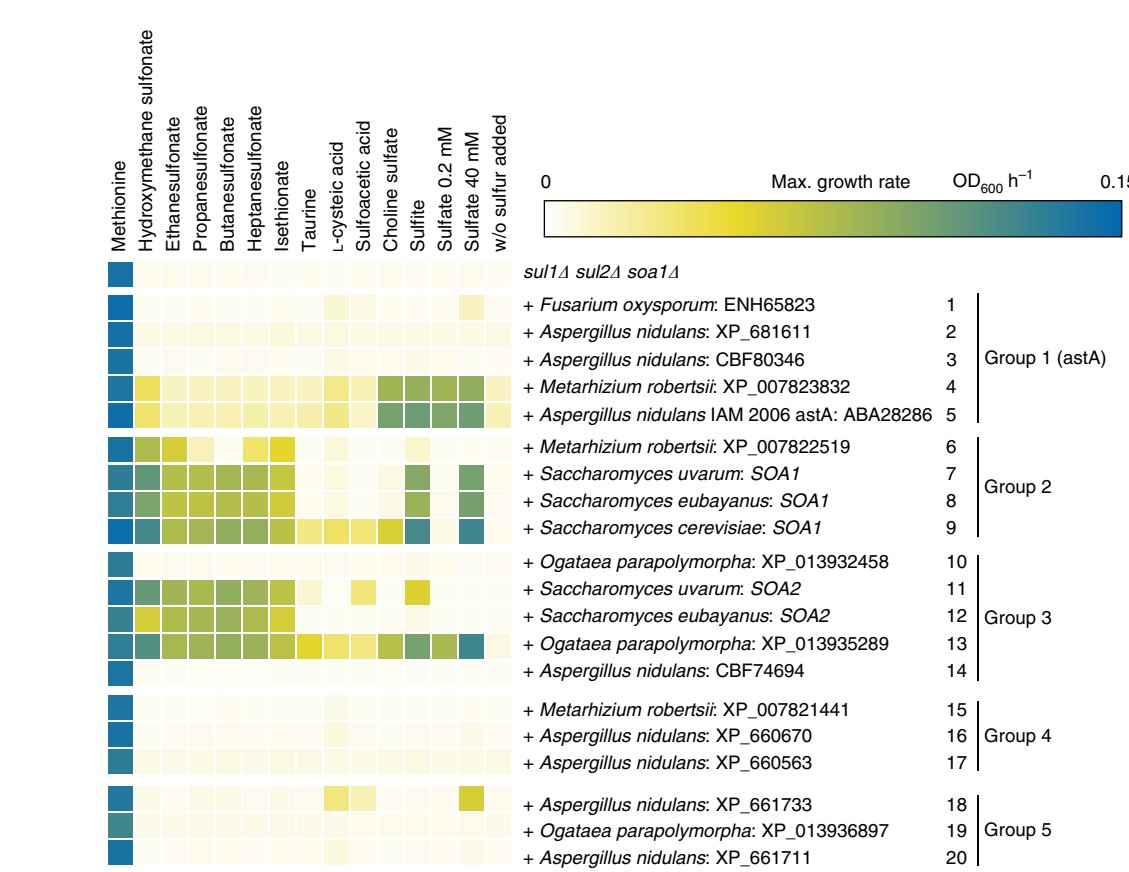

1, 2 and 3 there was good to very good growth with multiple sulfur sources for several members (Fig. 4b). Nearly all these members, however, belonged to the *Saccharomycetaceae*, with the orthologue from *Metarhizium robertsii* being the only exception. The latter showed an intermediate profile with good growth only on some of the sulfonates. The *M. robertsii* and *S. eubayanus* *SOA1* orthologues have the highest specificity for sulfonates of all carriers tested, whereas the *Ogatea parapolymorpha* orthologue was the most promiscuous carrier, transporting all 15 sulfur sources tested, even sulfate at low concentration (Fig. 4b). The members of the *astA* group that showed significant growth had a clear preference for inorganic sulfur sources, with choline sulfate being the only organic sulfur source supporting good growth. Good growth on sulfite correlates in all cases with good growth on high sulfate levels. Taurine, L-cysteic acid and sulfoacetic acid generally supported no or poor growth with all tested orthologues (Fig. 4b).

***Saccharomyces* species *SOA2*.** Besides *SOA1*, which was conserved in all *Saccharomyces sensu stricto* species in its location on chromosome 9 (except for *Schefflera arboricola*), we have discovered a new gene (*SOA2*) that appeared only in intact form in *S. uvarum* and *S. eubayanus* on chromosome 15. We also identified a *SOA2* open reading frame (ORF) in *S. arboricola* that was not previously described in the genome assembly. Thus, *S. uvarum*, *S. eubayanus* and *S. arboricola*, which split ~20 million years ago from the remaining *Saccharomyces* species[31], may encode two sulfonate transporters in their genome. The Soa1 and Soa2 transporters have a typical predicted 12-helix transmembrane structure with two cytosolic termini and a conserved 56–58-aa-long cytosolic loop separating two 6-helical bundles (Fig. 5a). The Soa2 gene products in *S. uvarum*, *S. eubayanus* and *S. arboricola* were highly similar to the corresponding parts of the products of the *S. cerevisiae* uncharacterized ORFs *YOL163W* and *YOL162W*, which are present in a similar position on chromosome 15 (except in *S. arboricola*). Eighty-two per cent and 73% of their residues were identical to those in the corresponding parts of *S. uvarum* Soa2, respectively. In *S. cerevisiae*, a large amino-terminal part of the Soa2 gene product is missing and a stop codon is present between the sequences encoding transmembrane domain 6 and 7 (Fig. 5a). As these genes are also highly upregulated under sulfur starvation conditions[18], it appears that *YOL163W* and *YOL162W* are inactive evolutionary leftovers of a functional *SOA2* gene. The relatively low sequence identity of 40–41% at the protein level between the paralogous genes *SOA1* and *SOA2*, and the recent loss of function of the *SOA2*-like ORFs (*YOL162W–YOL163W*) in *S. cerevisiae* in the group of *Saccharomyces sensu stricto* species that is estimated to have evolved 20 million years ago[31], suggest that they originate from a much more ancient gene duplication.

The *S. uvarum*, *S. eubayanus* and *S. arboricola* *SOA1* and *SOA2* genes may encode two sulfonate transporters that complement each other. Indeed, single deletion of *SOA1* or *SOA2* in *S. uvarum* resulted in a very similar sulfur compound utilization profile as the parental wild-type strain, whereas double deletion of *SOA1* and *SOA2* caused a strong reduction of growth with mainly the sulfonate compounds as the sole source of sulfur (Fig. 5b). Only growth with butanesulfonate and hydroxymethane sulfonate was just partially dependent on Soa1 and Soa2. Growth with sulfite, choline sulfate and sulfate was not dependent on Soa1 and Soa2.

**Adjacent genes syntenic with *SOA1*.** We also investigated the sequence similarity of the two adjacent genes on each side of *S. cerevisiae* *SOA1* (syntenic upstream and downstream). The *SDL1* and *NIT1* genes flanking *SOA1* upstream and downstream, respectively, are present as inactivated forms in the laboratory yeast S288C, due to the presence of a nonsense mutation in the middle of the genes, and we therefore used the wine yeast EC1118 full-length versions. None of the two genes was as conserved in fungi as *SOA1*. The product of the *NIT1* gene downstream of *SOA1* showed high sequence similarity to multiple predicted proteins from proteobacteria (Supplementary Data 3). Hence, the *NIT1* gene seems to have been transferred similarly as the *BDS1* sulfatase gene by horizontal gene transfer from proteobacteria[32]. The *BDS1* sulfatase gene is found syntenic with *SOA2* in *S. cerevisiae*, *S. uvarum*, *S. eubayanus* and *S. Arboricola*, and is responsible for their unique sulfatase activity[32]. *NIT1* and *SDL1* are highly upregulated when the yeast is supplied with less preferred nitrogen sources[33]. However, none of the yeast species tested with functional *NIT1* and *SDL1* genes could use the nitrogen in taurine or in glucosinolates (sinigrin), isothiocyanates (butyl and phenyl ethyl isothiocyanate), or ammonium sulfamate.

## Discussion
In the present study we have identified the last remaining transporter of sulfur compounds in the yeast *S. cerevisiae*. Previous research had shown that deletion of the two major sulfate transporter genes *SUL1* and *SUL2* did not prevent growth with high sulfate concentrations, nor with alternative sulfur sources such as sulfite and thiosulfate[16,20,21]. We have now shown that additional deletion of only one predicted transporter gene, encoded by *YIL166C* or *SOA1*, completely abolishes any residual growth of a *sul1Δ sul2Δ* strain with all sulfur sources, except for sulfur-containing amino acids, which are known to be transported by amino acid carriers[34,35]. As a consequence, the *sul1Δ sul2Δ soa1Δ* strain is an excellent tool for expression of heterologous candidate transporters of sulfur-containing compounds and determination of their substrate specificity and transport kinetics. The strain also allows to exclude any uptake of

**Figure 4 | Phylogenetic relationship and substrate preference of Soa1 fungal orthologues.** (**a**) Phylogenetic tree of alternative sulfate and sulfonate transporter orthologues, which were found with a BLASTP search against all non-redundant peptides and TBLASTN search against the *Saccharomyces* species not represented in the non-redundant peptide database. A protein tree was made with the program BEAST, using 70 million generations sampled every 5,000th generation, from which the maximum clade credibility tree is shown. The tree height was set to 1, with 0 being present day, measured from the root (1) to the tips (0) and 95% credibility intervals of the node heights (error bars) are only shown when the posterior probabilities were 0.95 and above. Dots with numbers correspond to the various fungal Soa1 orthologues tested for substrate preference after expression in the *S. cerevisiae sul1Δ sul2Δ soa1Δ* strain. The collapsed groups 4 and 5 are shown in full in Supplementary Data. (**b**) Substrate preference of SoA1 orthologues expressed in the *S. cerevisiae sul1Δ sul2Δ soa1Δ* strain. The NatMX cassette in the *SOA1* locus of the *sul1Δ sul2Δ soa1Δ* strain was replaced with gene sequences encoding fungal *SOA1* orthologues from different phylogenetic groups as indicated. The growth with sulfonates, choline sulfate, sulfite and sulfate (at 0.2 and 40 mM for high- and low-affinity transport) as sole sulfur source was assayed in liquid medium and growth with methionine was used as positive control. Growth curves were fitted to logistic growth models, except for two cultures of *S. cerevisiae* expressing *F. oxysporum* ENH65823, showing atypical growth dynamics, which were fitted to a smoothed spline in the linear growth phase. The *A. nidulans* predicted proteins 2 (XP_681611) and 3 (CBF80346) are potential splice variants derived from the same locus AN8342, although the existence of XP_681611 mRNA was rejected from the current gene model (www.aspergillusgenome.org).

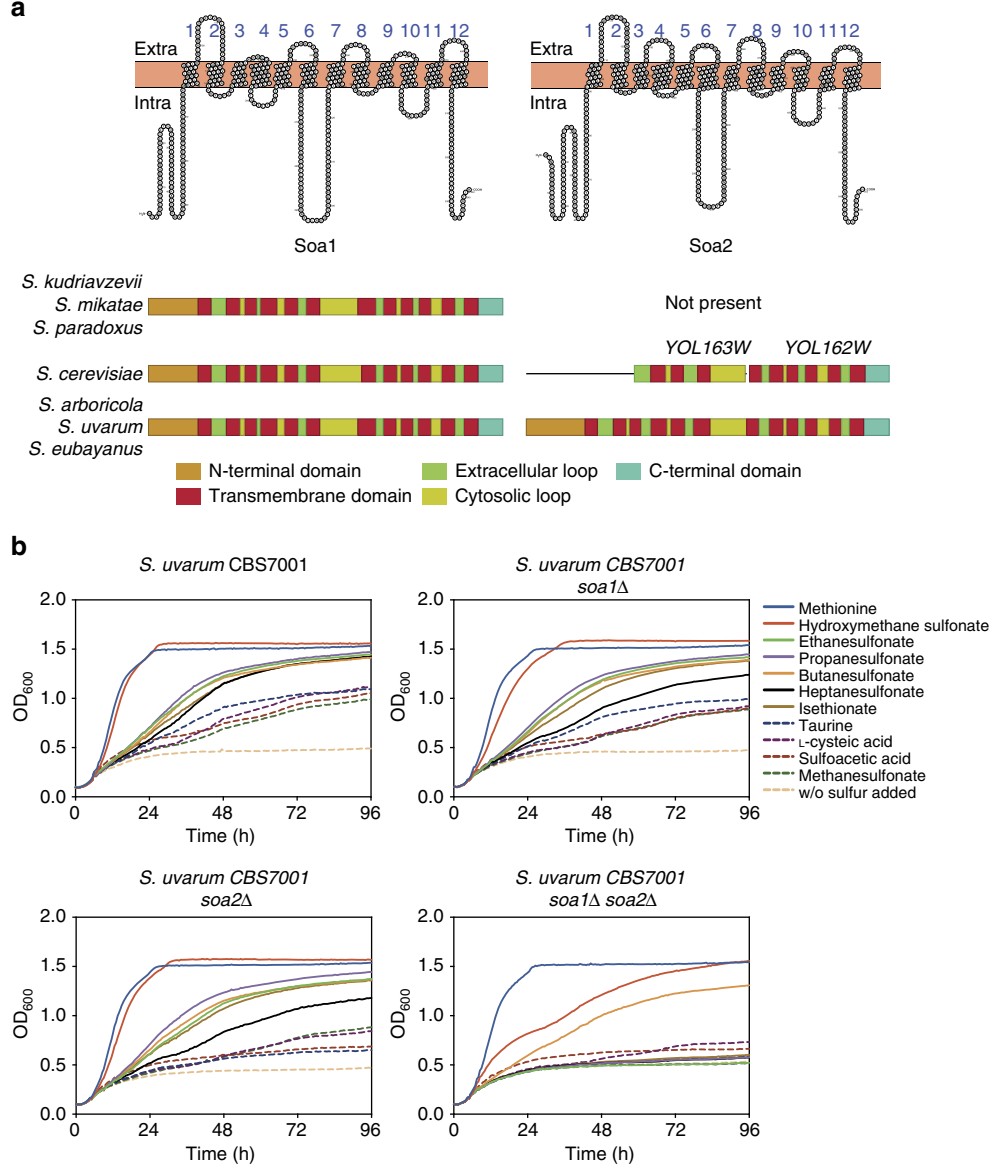

**Figure 5 | Comparison of *SOA1* and *SOA2* in *Saccharomyces* yeasts.** (**a**) Orthologous genes to *S. cerevisiae SOA1* were present in all the *Saccharomyces sensu stricto* species, whereas *SOA2* orthologues were only present in *S. arboricola*, *S. uvarum* and *S. eubayanus*. This is consistent with an early split in evolution since these species have the earliest split time of all *Saccharomyces* species. The previously uncharacterized *S. cerevisiae* genes *YOL163W* and *YOL162W* are split ORFs with high sequence similarity to the corresponding parts in *S. uvarum SOA2*. The topology and orientation of the *S. cerevisiae SOA1* and *S. uvarum SOA2* encoded transporters were predicted with TMPred and the predicted structure was plotted with Protter. (**b**) Substrate preference of *S. uvarum* Soa1 and Soa2. The *SOA1* and *SOA2* genes of *S. uvarum* were deleted in the diploid CBS7001 strain, the strain was sporulated and segregants containing either wild-type genes, or single or double deletions were selected. These were assayed for growth in liquid media at 25 °C with various sulfonates as sole source of sulfur. The results shown are the average for strains from two independent tetrads.

trace amounts of sulfur containing breakdown products from for instance sulfur-containing amino acid analogues.

We surprisingly show that the *SOA1* gene product functions as the main sulfonate transporter in *S. cerevisiae*. Growth of *S. cerevisiae* and other ascomycetes with sulfonates as the sole source of sulfur has previously been demonstrated and enzymes responsible for sulfonate breakdown were also identified[8,9]; however, the transporter(s) responsible for sulfonate uptake have remained unknown. Soa1 has broad substrate specificity taking up multiple types of sulfonates with medium affinity and likely to be using proton symport. Sulfonates are a major source of sulfur in certain soils. In forest soils, for instance, the majority of available sulfur is carbon bound and only 5% on average is present as inorganic sulfur[36,37]. A report on tropical forest soils shows that 78–97% of available sulfur is present under the form of sulfonates and 27–57% of the organic sulfur could be mobilized by the microbial population within 24 h (ref. 3), underscoring the physiological role of sulfonate uptake. Here we have demonstrated that the *S. cerevisiae* Soa1 sulfonate transporter is crucial for uptake of the naturally occurring L-cysteic acid, taurine, isethionate and a range of the other aliphatic sulfonates. Moreover, our study revealed that orthologues of this transporter are widely distributed and conserved in genetically distant fungi. For example, the endogenous transporters of the *O. parapolymorpha* and *M. robertsii* fungi have a similar capacity for sulfonate transport as Soa1. It seems that we have identified a family of organic sulfur transporters that are critical for fungal survival in ecological niches with low inorganic sulfur, from which fungi are

frequently isolated. On the other hand, our results with the closest Soa1 orthologue AstA from *A. nidulans*, expressed in the *S. cerevisiae sul1Δ sul2Δ soa1Δ* strain, showed that it functions mainly as a high-affinity sulfate transporter, whereas it supports only poor growth with sulfonates. The other orthologues of the Soa1 family for which we tested support of growth after expression in the *S. cerevisiae sul1Δ sul2Δ soa1Δ* strain displayed a diversity of substrate preferences for sulfur compounds, indicating that sequence conservation in the Soa1 family is not correlated with conservation of substrate preference. Finally, it has to be emphasized that expression of the heterologous transporters in *S. cerevisiae* may lead to structural modifications not occurring in the original host, which in principle could have an effect on substrate specificity and/or uptake kinetics.

Sulfonate metabolism has been investigated in great detail in bacteria[6,38]. The utilization of sulfonates in *Escherichia coli* resembles that of *S. cerevisiae*, which can degrade alkanesulfonates, L-cysteic acid and taurine, but not aromatic sulfonates[39], whereas other bacteria have an extended substrate range due to specific transport and degradation enzymes encoded on dedicated operons for, for example, *N*-acetyltaurine[40], isethionate[13], sulfoacetate[14], 4-toluenesulfonate[41] and aromatic sulfonates[42]. Some bacteria can use both the carbon and nitrogen in sulfonates to support growth and some can accomplish this even anaerobically[43–45]. Specific operons for sulfonate degradation have been found in the soil bacteria *C. necator* and *Comamonas testosteroni*, and for choline sulfate in the soil and gut bacterium *Bacillus subtilis*[46], suggesting the degradation of these sulfur compounds can provide specialist bacteria with an added fitness bonus by having an extended range of nutrients. In the present study, we show that transporters from *S. cerevisiae*, *S. uvarum*, *S. eubayanus*, *M. robertsii* and *O. parapolymorpha* have a broad substrate range for sulfonates, and show that the *S. cerevisiae* transporter Soa1 has medium affinity for isethionate. In contrast to our findings with the fungal transporters, which have rather broad substrate specificity and are likely to function as proton symporters, bacteria contain specific ATP-driven sulfonate and choline sulfate transporters[6,46].

The overlap in ecological niches between bacteria and fungi, and the fierce competition for scarce nutrients in nature, could be consistent with the existence of multiple specialized transporters. Except for *M. robertsii*, which is found in high titres in grassland soils that are rich in sulfate esters, consistent with the activity of the *M. robertsii AstA* gene (Fig. 4b), the yeast species investigated in our work live in widely varying environments, for example, bark, fruits, insects and soil, and can be considered generalists. Heterologous expression of the highly duplicated *SOA1* orthologues from the specialist plant pathogens *Fusarium oxysporum* and *V. longisporum* may reveal specific fungal high-affinity sulfonate transporters, which may provide a fitness advantage in their natural niche.

Our extensive phylogenetic analysis of fungal *SOA1* orthologues has led to several conclusions. First of all, we have learned that Soa1 belongs to a very large family of fungal transporters of which the substrate specificity was not well known. Second, that subdivision of the large Soa1 family based on sequence similarity fits approximately with subdivision based on substrate specificity. For instance, the astA subfamily mainly transports inorganic sulfur compounds and choline sulfate, whereas the transporters more closely related to Soa1 are in general also able to transport a range of sulfonates. Third, it has revealed the presence of many duplicated sequences, such as *SOA1* and *SOA2*, supporting the significance of 'non-classical' sulfur sources for growth and survival of fungi in nature. Fourth, we have found strong conservation of the Soa1 family in fungi, with members in all taxonomic divisions from the lower primitive groups to the highly evolved basidiomycetes. This again underscores the importance of these carriers and their substrates for the fungal kingdom.

Surprisingly, *S. cerevisiae* contains two partial genes, *YOL163W* and *YOL162W*, with sequence similarity to the corresponding parts of *SOA1* and which seem to be evolutionary leftovers of an ancient *SOA2* gene. Interestingly, this *SOA2* gene has been conserved in functional form in three species of the *Saccharomyces* genus, *S. uvarum*, *S. eubayanus* and *S. arboricola*, whereas in the other species, *S. kudriavzevii*, *S. paradoxus* and *S. mikatae*, *SOA2* has apparently been completely lost. This might be related to the role of sulfonates as source of sulfur in the ecological niches where the different species generally survive. *S. uvarum* and *S. eubayanus* prefer more cold-temperate climates and both have been found in association with trees[47]. However, the connection with sulfonate availability is not obvious. The same is true for the other species, as they have been found in a variety of different habitats, including soils and different parts of trees. Outside of the human-made environments, for instance, for the production of alcoholic beverages, *S. cerevisiae* and the other species have been isolated from bark and exudates of oak and pine trees[48–51], and nothofagus trees[52], and it was not until recently that a thorough characterization of wild *S. cerevisiae* strains was performed in isolated tropical forests in China, far away from human activity[53]. The genetic variability of the *S. cerevisiae* strains isolated in this study from the primeval tropical forests on the island of Hainan was as large as that of the entire population of *S. cerevisiae* strains isolated up till then. Hence, tropical forests may be a major original niche for *S. cerevisiae*[54]. Comparative research on the niches in which the different *Saccharomyces* species are found in these forests might reveal a link with the presence of sulfonates as sulfur sources and the functional conservation of the *SOA2*-encoded sulfonate transporter.

Our results show that Soa1 has a similar $K_s$ (half-velocity constant) for growth with sulfite as Sul1 and Sul2. Hence, Soa1 is a major sulfite carrier of *S. cerevisiae*. Sulfite is commonly used in the wine industry as an anti-contaminant and antioxidant, but its addition lowers ethanol yield and increases glycerol production by making a conjugate with acetaldehyde[55]. Wine strains often have enhanced tolerance to sulfite, which has been related to upregulation of the *SSU1* gene, encoding a sulfite efflux pump[23–25]. Our results suggest that inactivation of Soa1 by deletion of the *SOA1* gene might offer an additional approach to further enhance sulfite tolerance of wine yeast strains.

Another group of widely distributed organic sulfur compounds are sulfate esters, of which choline sulfate is well known to be important for sulfur turnover in the soil[1,2]. They are present in soils, being released by bacteria and taken up by roots of plants[56]. Sulfate esters have been proposed as a significant source of sulfate in soils through hydrolysis by microbial sulfatases[57]. However, sulfate esters can also serve as an intracellular source of sulfate. Many years ago, it was already demonstrated that *A. nidulans* could use choline sulfate as a source of sulfur, and that it was taken up by a different mechanism from that used for inorganic sulfate transport[58]. In related work, evidence was provided for a highly specific uptake system for choline sulfate[59]. *A. nidulans* is able to grow on a range of aliphatic and aromatic sulfate esters including the cyanogenic glucosinolate sinigrin[60]. Up to now, no fungal transporter for choline sulfate has been identified. Our experimental screening of multiple orthologues of the Soa1 fungal family has identified a functional choline sulfate transporter from *A. nidulans*, *M. robertsii* and *O. parapolymorha* (Fig. 4b). Together with *S. cerevisiae* Sul1, Sul2 and Soa1, these are choline sulfate transporters identified in fungi. However, none of these are specific for choline sulfate or sulfate esters in general, nor even for organic sulfur compounds. All six are also able to transport for instance sulfate and sulfite (Fig. 4).

In summary, our work has identified Soa1 as the last remaining inorganic sulfate transporter in *S. cerevisiae*, responsible for uptake of sulfate, sulfite and thiosulfate. Soa1 also functions as a sulfonate transporter. We have shown the usefulness of the *S. cerevisiae sul1Δ sul2Δ soa1Δ* strain for characterization of sulfur compound uptake of heterologous transporters by expressing 20 members of the fungal Soa1 family in this strain. This revealed a diversity of substrate preferences for sulfur-containing compounds, including other transporters for sulfonates and fungal transporters for a sulfate ester, that is, choline sulfate. Our findings underscore the ecological importance of these compounds as sources of sulfur in nature and suggest that there may be other ecologically relevant nutrients for which transporters have not been discovered yet in fungi.

## Methods

**Yeast strains.** The yeast strains used in this study are shown in Table 1.

**Sulfur compounds.** The sulfur compounds were of high-grade purity from Sigma-Aldrich or Merck. Choline sulfate was custom-made by ChiroBlock GmbH to minimum 95% purity and guaranteed to contain no detectable amounts of other sulfur compounds. The synthesis was carried out by incubating the choline bromide salt mixed with $Ag_2SO_4$ in ultra pure water and incubating for 2 weeks at $100\,°C$ with stirring. The liquid was crystallized repeatedly to obtain 95% purity without any detectable reagent compounds as followed by NMR and mass spectrometry.

**Growth conditions and sulfur starvation.** Cells were grown from OD 0.2 to OD 1 in yeast extract, peptone, dextrose (YPD) and starved for sulfur during 3 days in synthetic close-to-sulfur-free B medium (with ∼1.4 μM residual sulfate) composed according to Cherest *et al.*[61] and containing 4% glucose refreshed every day. Growth assays with the sulfur compounds as the sole source of sulfur were performed in the sulfur starvation medium, adjusted to pH 6.5 with hydrochloric acid (Sigma-Aldrich TraceSELECT Ultra) in 7 mM $K_2HPO_4$. The concentration of the supplied sulfur compounds was 200 μM, except where otherwise indicated. The pH profile of growth with sulfite was assayed in sulfur starvation medium buffered at the appropriate pH with 20 mM $K_2HPO_4$ and 20 mM malic acid. Cells were washed three times in sulfur starvation medium before being used for growth experiments.

**Yeast genetic modification.** A prototrophic *sul1Δ::KanMX sul2Δ::KanMX* strain was made by crossing the two single deletion strains from the BY deletion collection and subsequent backcrossing to S288c. The *YIL166C(SOA1)*, *YGR125W* and *YCR003C* genes were deleted by replacing the ORF with a split nourseothricin marker (NatMX) fused with 400–500 bp flanking regions to facilitate homologous recombination. LiAc transformation was done according to standard protocols. Transformation of *S. uvarum* was performed as for *S. cerevisiae*, except that heat shock was carried out for 1 h at $37\,°C$, and cells were grown at $25\,°C$. To obtain *SOA1* and *SOA2* single and double deletions in *S. uvarum*, the genes were first deleted in the homozygous diploid *S. uvarum* CBS7001-type strain with the KanMX and NatMX marker, respectively. Segregants were then obtained by sporulating the strain for 5 days at $23\,°C$ and subsequent tetrad dissection. The segregants were then tested for the resistance markers to select single and double deletions.

### Table 1 | Strains used in this study.

| Name | Strain background | Genotype | Mating type |
|---|---|---|---|
| *S. cerevisiae* strains | | | |
| BY4741 | Wild type | *sul1::KanMX* | MATa |
| BY4742 | Wild type | *sul2::KanMX* | MATalfa |
| S288c | Wild type | | MATa |
| SYH102 | S288C/BY | *soa1::NatMX* | MATalfa |
| SYH004 | S288C/BY | *sul1::KanMX sul2::KanMX* | MATalfa |
| SYH103 | S288C/BY | *sul1::KanMX sul2::KanMX soa1::NatMX* | MATalfa |
| SYH104 | S288C/BY | *sul1::KanMX sul2::KanMX soa1::NatMX*+Cas9-Ble | MATa |
| SYH90 | S288C/BY | *sul1::KanMX sul2::KanMX ygr125w::NatMX* | MATa |
| SYH91 | S288C/BY | *sul1::KanMX sul2::KanMX ypr003c::NatMX* | MATalfa |
| SYH70 | S288C/BY | *sul1::KanMX sul2::KanMX soa1::S. cerevisiae SOA1* (reinserted) | MATalfa |
| SYH71 | S288C/BY | *sul1::KanMX sul2::KanMX soa1::S. eubayanus SOA1* | MATalfa |
| SYH72 | S288C/BY | *sul1::KanMX sul2::KanMX soa1::S. uvarum SOA1* | MATalfa |
| SYH73 | S288C/BY | *sul1::KanMX sul2::KanMX soa1::M. robertsii XP_007822519* | MATalfa |
| SYH74 | S288C/BY | *sul1::KanMX sul2::KanMX soa1::A. nidulans CBF74694* | MATalfa |
| SYH75 | S288C/BY | *sul1::KanMX sul2::KanMX soa1::O. parapolymorpha XP_013932458* | MATalfa |
| SYH76 | S288C/BY | *sul1::KanMX sul2::KanMX soa1::S. eubayanus SOA2* | MATalfa |
| SYH77 | S288C/BY | *sul1::KanMX sul2::KanMX soa1::S. uvarum SOA2* | MATalfa |
| SYH78 | S288C/BY | *sul1::KanMX sul2::KanMX soa1::O. parapolymorpha XP_013932458* | MATalfa |
| SYH79 | S288C/BY | *sul1::KanMX sul2::KanMX soa1::A. nidulans IAM 2006 astA (ABA28286)* | MATalfa |
| SYH80 | S288C/BY | *sul1::KanMX sul2::KanMX soa1::M. robertsii XP_007823832* | MATalfa |
| SYH81 | S288C/BY | *sul1::KanMX sul2::KanMX soa1::A. nidulans CBF80346* | MATalfa |
| SYH82 | S288C/BY | *sul1::KanMX sul2::KanMX soa1::A. nidulans XP_681611* | MATalfa |
| SYH83 | S288C/BY | *sul1::KanMX sul2::KanMX soa1::F. oxysporum ENH65823* | MATalfa |
| SYH84 | S288C/BY | *sul1::KanMX sul2::KanMX soa1::M. robertsii XP_007821441* | MATalfa |
| SYH85 | S288C/BY | *sul1::KanMX sul2::KanMX soa1::A. nidulans XP_660670* | MATalfa |
| SYH86 | S288C/BY | *sul1::KanMX sul2::KanMX soa1::A. nidulans XP_660563* | MATalfa |
| SYH87 | S288C/BY | *sul1::KanMX sul2::KanMX soa1::A. nidulans XP_661733* | MATalfa |
| SYH88 | S288C/BY | *sul1::KanMX sul2::KanMX soa1::O. parapolymorpha XP_013936897* | MATalfa |
| SYH89 | S288C/BY | *sul1::KanMX sul2::KanMX soa1::A. nidulans XP_661711* | MATalfa |
| *S. uvarum* strains | | | |
| CBS7001 | Wild type | | MATa/alfa |
| SYH93 | CBS7001 | Tetrad C | MATa/alfa |
| SYH94 | CBS7001 | Tetrad C, *soa1::KanMX* | MATa/alfa |
| SYH95 | CBS7001 | Tetrad C, *soa2::NatMX* | MATa/alfa |
| SYH96 | CBS7001 | Tetrad C, *soa1::KanMX soa2::NatMX* | MATa/alfa |
| SYH97 | CBS7001 | Tetrad D | MATa/alfa |
| SYH98 | CBS7001 | Tetrad D, *soa1::KanMX* | MATa/alfa |
| SYH99 | CBS7001 | Tetrad D, *soa2::NatMX* | MATa/alfa |
| SYH100 | CBS7001 | Tetrad D, *soa1::KanMX soa2::NatMX* | MATa/alfa |

**CRISPR-mediated gene exchange.** Standard CRISPR/Cas9 technology was used[62]. The guide RNA plasmid and the Cas9 expression plasmid were based on the paper by DiCarlo et al.[63], with the auxotrophic marker changed for the antibiotic resistance marker as indicated and the cloning site modified for Gibson assembly. Putative fungal transporter genes were inserted into the soa1::NatMX locus by targeting the NatMX marker with the following guide RNA: 5′-TGTCCT CGACGGTCAGCCGG-3′ (PAM sites were present both in the seed sequence and the target site). The gRNA was designed for S. cerevisiae with DNA 2.0 CRISPR gRNA design tool and checked for potential off-targeting with BLAST. A single copy Cas9 constitutive expression plasmid with a phleomycin marker (BleMX) was transformed in the sul1Δ::KanMX sul2Δ::KanMX soa1Δ::NatMX strain. This strain was then transformed with a PCR-amplified fungal transporter gene together with a multicopy plasmid expressing a Nat-specific gRNA and a hygromycin selection marker (hphMX). Oligos with 60–70 bp SOA1 flanking sequences were used to PCR amplify genomic DNA from S. cerevisiae S288c, S. uvarum CBS7001, S. eubayanus CBS12357 and O. parapolymorpha DL-1. Genes encoding Soa1 transporter orthologues from A. nidulans, M. robertsii and F. oxysporum were codon optimized for expression in S. cerevisiae and synthesized by Gen9, Inc. (Cambridge, USA) and IDT (Leuven, Belgium). The transformed amplicons with these genes also contained 112 bp upstream and 84 bp downstream SOA1 flanking sequences. All orthologues expressed in S. cerevisae, either obtained by amplification from genomic DNA or by gene synthesis based on codon-optimized predicted protein sequences, have been submitted to GenBank with the following accession numbers: KX912914-KX912934.

**Sulfur compound uptake experiments.** Radioactively labelled [1-C14] isethionate, [1,2-C14] taurine and [S35] Na$_2$SO$_4$ were obtained from American Radiolabeled Chemicals, Inc. Unless otherwise mentioned, 40 μl of sulfur-starved cell suspension (120 mg wet weight ml$^{-1}$) buffered in 10 mM Bis-Tris pH 6.5 was pre-incubated for 10 min at 30 °C, after which 10 μl of radioactive stock solution was added to a total volume of 50 μl. The cells were incubated for 1 min for sulfate and 5 min for isethionate and taurine, and immediately separated on a glass microfibre filter and washed twice with 10 mM ice-cold unlabelled sulfur compound solution and once with cold demi water. The filter was then placed in a vial with addition of 5 ml of scintillation liquid and radioactivity was measured with a Hidex 300 SL liquid scintillation counter. The final specific activities of the radioactive stocks were 26,540 c.p.m. nmol$^{-1}$ (final concentration 200 μM) and 355 c.p.m. nmol$^{-1}$ (final concentration 40 mM) for sulfate, 7,390 c.p.m. nmol$^{-1}$ (for final concentration of 50 μM) and 771 c.p.m. nmol$^{-1}$ (for final concentration of 500 μM) for isethionate and 764 c.p.m. nmol$^{-1}$ (final concentration 3.6 mM) for taurine. For kinetic measurements, dilutions were made to final concentrations of 0.67–50 μM for sulfate (from a 200 μM stock), 10–500 μM for isethionate (10–50 μM from a 50 μM stock and 100–500 μM from a 500 μM stock) and 0.2–3.6 mM for taurine (from a 3.6 mM stock). The dilutions were confirmed to have a specific activity very similar to the original stocks. To determine the pH profile of transport, cells were buffered in 100 mM malic acid pH 4–6, Bis-Tris pH 6–7 or Tris pH 7–9. To abolish the proton gradient and test for H$^+$ symport, we incubated 30 μl sulfur-starved cell suspension (160 mg wet weight ml$^{-1}$) buffered in 10 mM Bis-Tris pH 6.5 with 10 μl of CCCP dissolved in methanol/water for 10 min at 30 °C before the transport measurement. The final concentration of methanol was kept constant at 1.5%, also in the negative control.

**Phylogenetic analysis.** A protein BLASTP (BLAST 2.2.29 + standalone) search was performed against the non-redundant protein database (last accessed October 2015) and subsequently filtered for a minimum of 40% identity in 90% of the sequence of each High Scoring Pair with an e-value of ≤ 1e − 40. The F. oxysporum ENH65823 sequence was selected for inclusion in the final tree from a group of plant pathogen fungal sequences, showing early divergence in a preliminary tree made with less stringent criteria. This protein was branching off with the astA high-affinity sulfate transporters and was also predicted to be the earliest diverging protein with Bayesian evolutionary analysis (Fig. 4a). However, the S. cerevisiae cells expressing the ENH65823 protein only showed very low residual growth with 40 mM sulfate (Fig. 4b and Supplementary Data 2) and further interpretation of which transport function was derived first therefore remains speculative. Orthologues from the Saccharomyces sensu stricto species S. uvarum, S. mikatae and S. paradoxus were added using the genomic sequence (from the Saccharomyces sensu stricto resources web site: www.saccharomycessensustricto.org) so that a total of 1,099 unique protein sequences was reached. To filter out non-functional, truncated proteins, as well as oversized proteins comprising the alignment score, a size window requirement was set at 350–650 aa. This finally yielded 1,011 unique protein sequences, which were clustered with BLASTCLUST (part of the BLAST 2.2.14 standalone package) based on 99% similarity within 90% of the sequence of each High Scoring Pair, so that a final number of 828 clusters was obtained (Supplementary Data 1). Next, a sequence from each cluster was selected and aligned with TCOFFEE (psi-coffee) and conserved blocks were extracted with Gblocks (0.91b), arriving at a total length of 296 aa, which were used to make the final gene tree with BEAST 2 (v. 2.3.1)[64]. The best fit was a Yule birth model with a uniform prior and eight gamma categories of heterogeneity. The effective sample size was > 1,200 for all variables. A chain of 70 million generations, sampled at 5,000 generation intervals, was used to make the maximum clade credibility tree

with a 12% burnin by TreeAnnotator. The results were validated with a separate chain. Only nodes with a posterior probability of ≥ 0.95 were annotated with 95% credible intervals. Taxonomic information was extracted from NCBI with Bioconductor and the tree was visualized with iTOL and FigTree v. 1.4.2.

**Determination of growth rates.** All growth experiments were carried out in duplicate, except for determination of half-velocity constants ($K_s$) for sulfite and thiosulfate, which was done in triplicate. Unless otherwise mentioned, the mean values of the cell densities were fitted to logistic and generalized logistic models (Richards), from which the maximal growth rates ($\mu$), maximal densities ($A$) and the lag phases ($\lambda$) were estimated with the R package GroFit[65]. In case of slow/residual growth ($\mu < 0.01$) the lag phase estimates are not reported. The cultures of S. cerevisiae expressing F. oxysporum ENH65823 with L-cysteic acid and 40 mM sulfate as the sole sulfur source displayed an atypical growth behaviour. Maximal growth rates for these atypical curves were fitted with a smoothed spline for the linear exponential growth phase between OD 0.2 and 0.4.

**Reproducibility of the results.** All experimental results were at least duplicated for confirmation.

**Data availability.** Accession codes for sequence data deposited at GenBank: KX912914-KX912934. All the remaining data supporting the findings of this study are available within the article and its Supplementary Information files, and from the corresponding author upon reasonable request.

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

## Acknowledgements

We thank the Carlsberg Research Laboratory, Copenhagen, Denmark, for providing part of the bioinformatics infrastructure. The phylogenetic calculations were carried out at the High Performance Computing (HPC) cluster of KU Leuven and we acknowledge the assistance of Martijn Oldenhof. We also acknowledge the expertise and help of Willy Verheyden with the radioactive uptake experiments and Nico Vangoethem for help with preparation of the figures. This work has been supported by grants from the Fund for Scientific Research–Flanders, Interuniversity Attraction Poles Network P7/40 and the Research Fund of the KU Leuven (Concerted Research Actions) to J.M.T.

## Author contributions

The experiments on sulfonates were designed by S.H. and J.M.T. The experiments on low-affinity sulfate transport were designed by H.K., S.H., G.V.Z. and J.M.T. The experiments on sulfite and pH dependence were designed by S.H., S.D.G., M.R.F.-M. and J.M.T. CRISPR oligos were designed by S.H. and M.R.F.-M. The phylogenetic analysis was designed, analysed and carried out by S.H. and S.L. The experimental work was carried out and analysed by S.H., H.K. and S.D.G. The manuscript was written by S.H. and J.M.T. and edited by the co-authors.

## Additional information

**Competing financial interests:** The authors declare no competing financial interests.

