## [Peer Review File · Nature Communications]

Reviewer #1 (Remarks to the Author)

Holt and colleagues describes the characterization of Soa1 as sulfur compound transporter in *Saccharomyces cerevisiae*. The authors indicate that Soa1 is the last remaining sulfate transporter, which also functions sulfonate transporter; and they also construct a relevant null strain. The manuscript is well written and the data are clearly presented, but the paper is rather long and the amount of mechanistic insight is limited. The paper is merely a description of sulfur compound transport in yeast and other fungi, and I feel that the paper is more suitable for a specialized yeast or microbiology journal. Below some specific comments.

Comments

1. Please refer to figure subpanels rather than e.g. Figure x
2. Lines 165-167: What is means meant with "... major carriers for high levels of sulfate"? Low affinity, high capacity transporters. Where do I find the relevant information in Figure 3? Is the 40 mM correct or should it be 40 microM?
3. Lines 179-182: What is the basis for the conclusion that AstA is 12x more efficient than Soa1? The rates seem to differ 80,000 fold. Without normalizing the expression levels at the plasma membrane, one cannot conclude much about the efficiency of transport.
4. The CCCP treatment is consistent with transporters functioning as proton symporters, but most transporters are also inactivated when the internal pH drops below 7 and this should be taken into account when analyzing the effects of protonophores.
5. What do we learn from the phylogeny about the transporters?
6. I would present the data of Fig.5B as growth rates in a table or bar diagram and move the growth curves to the supplement. Similarly, the other growth curves/data (Fig.1 and 2) might be better presented when growth rates and final ODs are extracted.
7. I personally don't like terminology like ".. we have discovered a novel gene (SOA2)..". (line 263), "...the first transporter for ...". (lines 407-414), and other similar statements elsewhere in the text. What is novel, what has been discovered, what is first; I would tone down the description and not oversell the work.
8. Line 385 and elsewhere. "Our results show that Soa1 has similar affinity...". I would indicate "Michaelis constant of transport" as affinity is readily confused with binding affinity and the description is not very precise.
9. I would shorten the sections on phylogeny and orthologues (both in the R and D sections) as the work is rather descriptive, and the lengthy description reduces the contents/length ratio.
10. Some of the curves in Fig. 1E and 1G are sigmoidal and suggest cooperativity. What is the explanation for this finding?

Reviewer #2 (Remarks to the Author)

This is a thorough, interesting study demonstrating that the *S. cerevisiae* ORF YIL166C is the main transporter for sulfonates. Sulfonates represent a major form of sulfur in soils, but the newly identified transporter is the first known sulfonate transporter in fungi. The study is therefore original and of interest. The approach is valid, and the conclusions are supported by the data. The manuscript is clearly written.

Some relatively minor points:

1. Line 430. For the sulfite growth experiments, non-enzymatic oxidation of sulfite to sulfate would progressively lower the sulfite concentration. Do the authors have information on the extent of sulfite oxidation under the conditions used in these studies?
2. Figure 2B. It is hard to distinguish all the colors used for the growth curves in different S sources, especially the dark blues. Dotted or dashed lines would help.
3. Figure 3, top. The panel showing the pH dependence of isethionate transport indicates that the transport associated with SOA1 (difference between WT and *soa1Δ*) declines as the pH is lowered from 7 to 5. The authors propose that the SOA1 gene product is a proton-coupled transporter. There should be some comment on the pH dependence of transport and whether other known proton-coupled transporters have a similar dependence on pH.

Very minor:

Lines 119-120. It is probably not necessary to have 3 significant figures for the Ks data.

Paragraph beginning line 131. The sulfite efflux transporter SSU1 is mentioned later in the manuscript, but it could also be mentioned here.

Line 159. Better to say "... indicating that neither SUL1 nor SUL2..."?

Lines 172-174 and 435-437. Although the method is widely used, it would be good to include a reference for the CRISPR targeting of the nourseothricin marker.

Lines 353-355. A reference is needed for the bacterial ATP-driven transporters.

Line 394. A reference is needed for statement that sulfate esters are taken up by plant roots.

Lines 420-423. Is there any information on the impurity in the 95% pure choline sulfate?

Reviewer #3 (Remarks to the Author)

The manuscript "SOA1 encodes the major sulfonate transporter in *Saccharomyces cerevisiae* and its fungal family shows considerable substrate diversity for sulfur compounds" describes a well-rounded set of experiments involving the identification of the first fungal transporter Soa1 for sulfonates and choline sulfate as sources of sulfur for *S. cerevisiae* and other fungi. Moreover, Soa1 is the last unraveled transporter of sulfur compounds in the yeasts.

Authors characterized multiple members with distinct substrate specificity in the large Dal5 family of fungal Soa1 homologues which display diversity of substrate preferences for oxidized sulfur compounds.

In my opinion, this work will be of general interest to the field of sulfur and transporters and suitable for Nature communications.

However, I suggest the following changes, which should improve manuscript

Major revisions:

Fig 1A would look better with additional wild type control, e.g. triple transformant [sul1+ sul2+ soa1+] or at least recipient WT strain.

Fig 1B Δ soa mutant saturation kinetics is actually a sum of Sul1 and Sul2 transporter activities. Since each of them has a different Michaelis constant, it would be more clear to split their saturation kinetics (separate Δ sul1 Δ soa1 from Δ sul2 Δ soa1) or at least mention about it somewhere in text.

Michaelis constant of sulfate transporters – Tweddle and Segel (1970) estimated K_T for *A. nidulans* sulfate permease as 75 μ M, which is one order far from constant calculated for yeast Sul1p and Sul2p. Similar order of K_T was estimated for *A. nidulans* AstA as 85 μ M and for two *Fusarium* AstAs c.a. 35-37 μ M (Pilsyk et al. 2015). What is the origin of such differences comparing to your results? Different method of assay? very short vs. long term of uptake? weight of wet vs. dry mycelia? May be due to heterologous expression and posttranslational modifications, for instance, glycosylation (like in mammalian OAT transporter). How did You stop uptake reaction before wash-out and how precisely did You wash-out residual radioactivity from the periplasmic space in yeast cells? Please, discuss this, present calculations more precisely and show data of sulfate uptake kinetics in figure at least as Lineweaver-Burk or Hanes-Woolf plot.

Growth of yeast strain expressing Dal5 homologs in various sulfur sources – *A. nidulans* XP_681611 and CBF80346 refer to same AN8342 locus, thus you have 19 tested transporters, not 20.

Minor revisions:

Significant abuse of word "ortholog". Protein may be named orthologous when enzymatic function was annotated experimentally with high similarity to another enzyme. Eventually, we may suppose similar function when proteins have high identity between each other (with e-value 0.0).

Otherwise better use word homolog, it is more safe.

As an osmophile, yeasts abundantly occupy ecological niche of leaf surface, fruits and in milk. Do You have any idea which sulfonates may be found in these environments?

Michaelis constant for transport (K_T). Since transporters do not modify substrate as enzymes, it is unprecise to use K_m symbol. Better use Greek tau letter to describe transportation.

In text and in figures: may be better use name of loci instead of accession numbers? e.g. A.

nidulans XP_681611 and CBF80346 refer to one same AN8342 locus.

Line 20 – inorganic (oxidized) sulfur anions rather than inorganic sulfate (sulfite and thiosulfate are not sulfate).

Line 167 – figure 3 is not clear to understand which subfigure is related to sulfate uptake.

Line 188, 534 – Unclear. Is it a gene tree or rather protein tree with protein sequences used?

Line 235 – *Metarhizium* is a filamentous fungus, not a yeast.

Line 237 – AstA (fourth capital letter refer to no locus).

Line 286 – butanesulfonate and hydroxymethane sulfonate

Line 390 – weak hypothesis. Even under repression, traces of Soa might be lethal in sulfite-containing wine. To prevent sulfite uptake, it would be better for yeast to loss Soa.

Line 477 – have You checked a negative control how 1.5% MeOH affect on H⁺ decoupling?

Line 506, 547 – means yeast expressing *F. oxysporum* gene, not *Fusarium* itself

Table 1 – I assume that should be *S. uvarum* Δ soa1/2 mutants instead of Δ sul1/2

Fig.3 miss slope in taurine and sulfate (SOA) transport plots. In many plots miss standard error bars

Fig.1 and 5 – w/o sulfur added (without slash, it resembles W0, the yeast minimal medium)

Answers to the comments of the reviewers

Reviewer #1

Comments

1. Please refer to figure subpanels rather than e.g. Figure x

This has now been done also for Fig. 3, so that for all figures we now refer to subpanels. Hence, also for Fig. 3 it should now be easy to navigate through the panels.

2. Lines 165-167: What is means meant with "... major carriers for high levels of sulfate"? Low affinity, high capacity transporters.

We have improved the text for clarity:

'Determination of the uptake of a high level of sulfate (40 mM) in the *sul1Δ sul2Δ* and *sul1Δ sul2Δ soa1Δ* strains indicated that the three carriers are responsible for all the uptake of high sulfate, while Sul1 and Sul2 are the major high-capacity carriers for high levels of sulfate, since their deletion strongly curtails the uptake of 40 mM sulfate (Fig. 3C).'

Where do I find the relevant information in Figure 3?

We have now referred to Fig. 3C.

Is the 40 mM correct or should it be 40 microM?

The 40 mM is indeed correct, since it concerns very-low-affinity uptake of sulfate.

3. Lines 179-182: What is the basis for the conclusion that AstA is 12x more efficient than Soa1? The rates seem to differ 80,000 fold.

We are sorry for an error in the text: $81.8 \pm 3.6 \mu\text{mol}\cdot\text{min}^{-1}\cdot\text{mg}^{-1}$ cell dry weight should have been $81.8 \pm 3.6 \text{pmol}\cdot\text{min}^{-1}\cdot\text{mg}^{-1}$ cell dry weight. This has now been corrected.

Without normalizing the expression levels at the plasma membrane, one cannot conclude much about the efficiency of transport.

We agree with the referee and we have made the description more precise: This comparison assumes similar expression levels at the plasma membrane. This cannot be guaranteed in spite of the fact that the *astA* protein sequence was codon optimized for expression in *S. cerevisiae* and engineered into the *soa1* locus ensuring use of the same promoter and terminator and at the same chromosomal position. We have added this in the text.

4. The CCCP treatment is consistent with transporters functioning as proton symporters, but most transporters are also inactivated when the internal pH drops below 7 and this should be taken into account when analyzing the effects of protonophores.

We agree with the referee that lower intracellular pH can counteract uptake of external molecules by symport, although we think that complete inactivation of the uptake at an internal pH lower than 7 is somewhat overstated. For instance, yeast cells can proliferate under anaerobic conditions at external pH of 4 and even below. Under these conditions, the cells will certainly not be able to maintain an intracellular pH above 7, and maybe not even above 6. And still they must have very active proton symport under such conditions. We should not forget that uptake of molecules by proton symport is driven both by the pH gradient and by the membrane potential.

We have deliberately used the cautionary statement: 'This is consistent with the transporters functioning as H⁺-symporters.' rather than using statements like 'this indicates' or 'this suggests' H⁺-symport.

5. What do we learn from the phylogeny about the transporters?

We have added the following explanation in the discussion section:

First of all, we have learned that Soa1 belongs to a very large family of fungal transporters of which the substrate specificity was not well known. Second, that subdivision of the large Soa1 family based on sequence similarity fits approximately with subdivision based on substrate specificity. For instance, the ASTA subfamily mainly transports inorganic sulfur compounds and choline sulfate, while the transporters more closely related to Soa1 are in general also able to transport a range of sulfonates. Third, it has revealed the presence of many duplicated sequences, such as SOA1 and SOA2, supporting the importance of 'non-classical' sulfur sources for growth and survival of fungi in nature. Fourth, we have found strong conservation of the SOA1 family in fungi, with members in all taxonomic divisions from the lower primitive groups to the highly evolved basidiomycetes. This again underscores the importance of these carriers and their substrates for the fungal kingdom.

6. I would present the data of Fig.5B as growth rates in a table or bar diagram and move the growth curves to the supplement. Similarly, the other growth curves/data (Fig.1 and 2) might be better presented when growth rates and final ODs are extracted.

We feel somewhat reluctant to implement these changes. The main purpose of the figures is to show what sulfur compounds can support growth and not what the precise growth rate and final OD are, since the latter are very much dependent on the precise conditions used. In addition, the lag phase is also variable and this type of variation in the growth curves is clearer with a visual representation. For Fig. 5B there is an additional argument that *S. uvarum* grows more slowly than *S. cerevisiae* and the growth differences are thus more clear when the whole growth curves are shown.

7. I personally don't like terminology like "... we have discovered a novel gene (SOA2).." (line 263), "...the first transporter for ..." (lines 407-414), and other similar statements elsewhere in the text. What is novel, what has been discovered, what is first; I would tone down the description and not oversell the work.

We agree to some extent with the referee. We certainly do not want to brag about our results. We have changed: 'we have discovered a novel gene (SOA2)' into 'we have discovered a new gene (SOA2)' because SOA2 can no longer be considered as 'novel' after the discovery of SOA1.

On the other hand, we consider Soa1 truly as a novel transporter because it is indeed the first fungal transporter discovered for sulfonates. (By the way, multiple titles in the reference list contain the word 'novel'.) We feel that this is important information for the reader.

In addition, the purpose of these statements can also be purely scientific, i.e. to provide interesting scientific information to the reader. When we mention for instance that Soa1 is the first sulfonate transporter discovered in fungi, this raises questions as to why it took so long to discover a sulfonate transporter, it also hints that more of these transporters may wait to be discovered (which we actually demonstrated in our paper) and that there may be other ecologically relevant nutrients for which transporters have not been discovered yet.

We have also taken great care to cite all relevant previous research, acknowledging the work of other scientists in this field.

8. Line 385 and elsewhere. "Our results show that Soa1 has similar affinity...". I would indicate "Michaelis constant of transport" as affinity is readily confused with binding affinity and the description is not very precise.

We did not really measure the Michaelis constant of transport for sulfite, but rather the Ks (half-velocity constant) for growth with sulfite as sole source of sulfur. We have improved the text as follows:

'Our results show that Soa1 has a similar Ks (half-velocity constant) for growth with sulfite as Sul1 and Sul2.'

9. I would shorten the sections on phylogeny and orthologues (both in the R and D sections) as the work is rather descriptive, and the lengthy description reduces the contents/length ratio.

We agree with the referee and we have shortened the sections on the 'Bayesian phylogeny of fungal orthologs' as well as the section on 'Expression of orthologs encoding putative sulfonate transporters in *S. cerevisiae*.'

10. Some of the curves in Fig. 1E and 1G are sigmoidal and suggest cooperativity. What is the explanation for this finding?

With active sul1 and sul2 transporters, some transport of trace amounts ($< \sim 1.4 \mu\text{M}$) of sulfate from the media (discussed in the text in relation with the results of Fig.4B) can be used for growth, in which case there is residual growth even without added sulfur source. This is accounted for by adding a constant to the Monod equation ($\mu = \mu_{\text{max}} \cdot [\text{S}] / (\text{Ks} + [\text{S}] + \text{constant})$). In the case of low affinity transport of sulfate by Soa1, the transport rate at low concentrations is not high enough to support growth before addition of 5 mM sulfate. Thus, the curve becomes sigmoidal because the minimal concentration needed for cell growth is lower than what can be transported.

Reviewer #2:

This is a thorough, interesting study demonstrating that the *S. cerevisiae* ORF YIL166C is the main transporter for sulfonates. Sulfonates represent a major form of sulfur in soils, but the newly identified transporter is the first known sulfonate transporter in fungi. The study is therefore original and of interest. The approach is valid, and the conclusions are supported by the data. The manuscript is clearly written.

Some relatively minor points:

1. Line 430. For the sulfite growth experiments, non-enzymatic oxidation of sulfite to sulfate would progressively lower the sulfite concentration. Do the authors have information on the extent of sulfite oxidation under the conditions used in these studies?

We do not think that sulfite is significantly oxidized to sulfate under our conditions, i.e. during the 72 hours of the growth experiment in malate-K₂HPO₄-buffered cells at pH 3.5-7. If this would be the case, it should compromise the growth of the sul1 sul2 deletion strain with low levels of sulfite,

because the remaining Soa1 transporter has only very low affinity for sulfate. We have not observed this in the growth experiments. Moreover, from the literature we know that the oxidation of sulfite to sulfate is only significant in alkaline conditions with metal ion catalysts, which are conditions that are very different from our incubation conditions.

2. Figure 2B. It is hard to distinguish all the colors used for the growth curves in different S sources, especially the dark blues. Dotted or dashed lines would help.

We agree with the referee and we have changed some lines into dotted and dashed lines. In addition, we have made the colored lines in the legend next to the graphs much thicker.

3. Figure 3, top. The panel showing the pH dependence of isethionate transport indicates that the transport associated with SOA1 (difference between WT and soa1 Δ) declines as the pH is lowered from 7 to 5. The authors propose that the SOA1 gene product is a proton-coupled transporter. There should be some comment on the pH dependence of transport and whether other known proton-coupled transporters have a similar dependence on pH.

We agree with the referee that faster transport is expected at low pH for a proton symporter. On the other hand, the co-substrate may be limiting for the rate of uptake at lower pH and as is true for all enzymes and transporters the activity is dependent on the protein structure, which is compromised at extreme pH values with a reduction in activity as a consequence.

The interpretation is also compromised by the interference with diffusion of sulfite at low pH.

We have added the following comment in the text:

Isethionate uptake by Soa1 drops from pH 6 to pH 5, which likely reflects a regular pH effect on protein structure/activity, similar to the pH profile of sulfate uptake, which does not suffer from interference with diffusion (see further, Fig. 3D).

Very minor:

Lines 119-120. It is probably not necessary to have 3 significant figures for the Ks data.

We agree that it may not be necessary, but it is more precise. The values reported are consistent with the calculated error, and the other reported kinetic constants for the transport measurements.

Paragraph beginning line 131. The sulfite efflux transporter SSU1 is mentioned later in the manuscript, but it could also be mentioned here.

We have added the following sentence:

'Sulfite can be exported by the Ssu1 efflux pump(Goto-Yamamoto et al., 1998; Nardi, Corich, Giacomini, & Blondin, 2010; Park & Bakalinsky, 2000).'

Line 159. Better to say "... indicating that neither SUL1 nor SUL2..."?
The text has been corrected.

Lines 172-174 and 435-437. Although the method is widely used, it would be good to include a reference for the CRISPR targeting of the nourseothricin marker.

We have added a general reference to the CRISPR/Cas9 methodology:

'Standard CRISPR/Cas9 technology was used(Ran et al., 2013).'

We designed the gRNA ourselves so this is the first report of its usage. The backbone of the plasmids is coming from DiCarlo et al. (2013), which is referenced in the paper. We have added a sentence about the design of the gRNA:

'The gRNA was designed for *S. cerevisiae* with DNA 2.0 CRISPR gRNA design tool and checked for potential off-targeting with BLAST.'

Lines 353-355. A reference is needed for the bacterial ATP-driven transporters.

We have added two relevant references.

Line 394. A reference is needed for statement that sulfate esters are taken up by plant roots.

We have added a relevant reference.

Lines 420-423. Is there any information on the impurity in the 95% pure choline sulfate?

We have improved the description:

'Choline sulfate was custom made by ChiroBlock GmbH to minimum 95% purity and guaranteed to contain no detectable amounts of other sulfur compounds.'

There is no further information about the impurities, except that they likely concern the solvents used (water and ethanol).

Reviewer #3 (Remarks to the Author):

The manuscript "SOA1 encodes the major sulfonate transporter in *Saccharomyces cerevisiae* and its fungal family shows considerable substrate diversity for sulfur compounds" describes a well-rounded set of experiments involving the identification of the first fungal transporter Soa1 for sulfonates and choline sulfate as sources of sulfur for *S. cerevisiae* and other fungi. Moreover, Soa1 is the last unraveled transporter of sulfur compounds in the yeasts.

Authors characterized multiple members with distinct substrate specificity in the large Dal5 family of fungal Soa1 homologues which display diversity of substrate preferences for oxidized sulfur compounds.

In my opinion, this work will be of general interest to the field of sulfur and transporters and suitable for Nature communications.

However, I suggest the following changes, which should improve manuscript

Major revisions:

Fig 1A would look better with additional wild type control, e.g. triple transformant [sul1+ sul2+ soa1+] or at least recipient WT strain.

We have added the growth results for the wild type strain in the same media on top. (This has to be done on different plates from the deletion strains because of interference with the growth of the deletion strains by an unidentified sulfur compound (possibly H₂S) excreted by the wild type strain.)

Fig 1B Δ soa mutant saturation kinetics is actually a sum of Sul1 and Sul2 transporter activities. Since each of them has a different Michaelis constant, it would be more clear to split their saturation kinetics (separate Δ sul1 Δ soa1 from Δ sul2 Δ soa1) or at least mention about it somewhere in text.

The kinetics of Sul1 and Sul2 have been reported in detail previously. Because of this reason and because the characterization of Soa1 was the main goal of the present paper, we have not included the soa1 Δ sul1 Δ and so1 Δ sul2 Δ strains in the experiment of Fig. 1B, C, D.

As requested by the referee, we have added the following comment in the text:

'In case of the soa1 Δ strain the K_s value is the result of the transporter kinetics of both Sul1 and Sul2.'

Michaelis constant of sulfate transporters - Tweddle and Segel (1970) estimated K_t for *A. nidulans* sulfate permease as 75 μ M, which is one order far from constant calculated for yeast Sul1p and Sul2p. Similar order of K_t was estimated for *A. nidulans* AstA as 85 μ M and for two *Fusarium* AstAs c.a. 35-37 μ M (Pilsyk et al. 2015). What is the origin of such differences comparing to your results? Different method of assay? very short vs. long term of uptake? weight of wet vs. dry mycelia? May be due to heterologous expression and posttranslational modifications, for instance, glycosylation (like in mammalian OAT transporter). How did You stop uptake reaction before wash-out and how precisely did You wash-out residual radioactivity from the periplasmic space in yeast cells? Please, discuss this, present calculations more precisely and show data of sulfate uptake kinetics in figure at least as Lineweaver-Burk or Hanes-Woolf plot.

We thank the reviewer for bringing these papers to our attention and we have carefully examined these studies.

Tweddle and Segel (1970) is an early study performed at a time when there were no genome sequences available and no precise knowledge of the number and type of transporters. These authors used mycelia of *Penicillium* and *Aspergillus* in rich media causing low uptake activity of sulfur compounds and in sulfur-limiting and starvation media causing very high uptake of sulfur compounds. They also used assays with short (2 min) uptake time. The K_m value of 75 μ M for sulfate uptake was determined with sulfur-limited cells. Sulfur limitation or starvation is well known to cause induction of high-affinity transporters. In addition, it is not known what transporter or transporters may be responsible for this high-affinity transport and thus we have no idea whether these transporters are the same as those used in our work. The referee correctly points out that the heterologous expression in the *S. cerevisiae* host cells may affect the structure of transporters and thus also the kinetics of their transport activity.

The Pilsyk et al. paper (2015), on the other hand, is a very recent study. However, in this study the authors only investigated sulfate assimilation into TCA-precipitable material over long time periods of 30 and 60 min, and not short-term sulfate uptake for assaying transport activity alone. In long-term sulfate assimilation assays, the transport step is not necessarily the rate-limiting step and the incorporation of sulfate into organic matter is certainly influenced by metabolic reaction equilibria, regulatory mechanisms operating on the enzymatic pathways and feedback-inhibition mechanisms operating on the transporters. Hence, the authors did not determine K_m values of transport but, as the referee also indicates, K_T values of sulfate assimilation.

Both studies have also used sulfate concentrations above the reported K_m or K_T values, which cannot provide an accurate determination of these parameters.

The difference in the affinity constant cannot be due to the cell mass as it should be a constant if the experimental conditions are OK. We have used a very short incubation time (1 min) to avoid time-dependent saturation and we have implemented all appropriate controls. To measure the K_m of *astA*, we diluted a radioactive stock of sulfate from 200 μM to the range of 1-40 μM used for the transport assay. All strains constructed were confirmed by sequencing and the experiments were appropriately repeated. We therefore believe that our measurements of the Michaelis-Menten constants are reliable. We have also chosen to show the raw data together with the fitted kinetics, so that readers can trace back all conclusions to the original data. Our results clearly fit Michaelis-Menten kinetics, and to make this even more convincing, we have now also added the R^2 values for *Soa1* and *ASTA* into the graphs.

We have also added in the Methods section the concentrations of the stocks made, and how they were diluted:

'The final specific activities of the radioactive stocks were 26540 Ci/nmol (final concentration 200 μM) for sulfate, 7390 Ci/nmol (for final concentration of 50 μM) and 771 Ci/nmol (for final concentration of 500 μM) for isethionate, and 764 Ci/nmol (final concentration 3.6 mM) for taurine. For kinetic measurements dilutions were made to final concentrations of 0.67-50 μM for sulfate (from a 200 μM stock), 10-500 μM for isethionate (10-50 μM from a 50 μM stock and 100-500 μM from a 500 μM stock), and 0.2-3.6 mM for taurine (from a 3.6 mM stock). The dilutions were confirmed to have a specific activity very similar to the original stocks.'

We have now referred to these two papers in the Results section:

'There is very little information on kinetic characteristics of sulfur compound transporters in fungi. An early paper by Tweedie and Segel (Tweedie & Segel, 1970) reports a K_m value of 75 μM for sulfate uptake in sulfur-limited *Aspergillus* cells but it is unclear what transporters were responsible for this uptake. A recent paper by Pilsyk et al. (Pilsyk, Natorff, Gawinska-Urbanowicz, & Kruszewska, 2015) has determined the kinetics of long-term sulfate assimilation into cellular organic matter and it is unclear how the kinetic constants determined relate to those of sulfate transport as measured in very short-term uptake experiments.'

We have also added several cautionary statements:

In Results section:

'Although the *astA* gene from *A. nidulans* was codon-optimized and inserted in the original *SOA1* locus, we cannot exclude that the protein lacks structural modifications happening in the original host or undergoes new modifications in *S. cerevisiae* which may alter its kinetics.'

In Discussion section:

'Finally, it has to be emphasized that expression of the heterologous transporters in *S. cerevisiae* may lead to structural modifications not occurring in the original host, which in principle could have an effect on substrate specificity and/or uptake kinetics.'

We have added Hanes-Woolf plots as insets in Fig. 3A and Fig. 3D, as requested by the referee.

Growth of yeast strain expressing Dal5 homologs in various sulfur sources – A. nidulans XP_681611 and CBF80346 refer to same AN8342 locus, thus you have 19 tested transporters, not 20.

The reason for the 20 transporters instead of 19 is that there are two splice variants: XP_681611 and CBF80346.

Had the proteins been identical they would have been clustered together (99% ID in the top HSP) and would be represented in one branch of the tree. However, the predicted protein sequences are 100% identical in 417 of the amino acid positions (456 aa for CBF80346 and 481 aa for XP_681611), and they are encoded from the same loci. The XP_681611 is the original predicted spliced variant (loci EAA66904), while the CBF80346 is predicted from the Aspergillus genome annotation update in 2008 (Wortman et al., 2009). We used the NetAspGene tool (Wang, Ussery, & Brunak, 2009) for identifying potential alternative splice sites in the AN83242.2 gene, and found 5 introns 2 which were not predicted with high confidence (<0.95). These two potential alternative spliceforms correspond to the CBF80346 and XP_681611 proteins, respectively. As they differed in 75 of amino acids, we chose to express both predicted variants.

We have now indicated in the Fig. 4B legend that XP_681611 and CBF80346 are differently predicted splice variants derived from the same locus AN8342.

Minor revisions:

Significant abuse of word "ortholog". Protein may be named orthologous when enzymatic function was annotated experimentally with high similarity to another enzyme. Eventually, we may suppose similar function when proteins have high identity between each other (with e-value 0.0). Otherwise better use word homolog, it is more safe.

The definition of 'ortholog' is a homolog in the same species, while a 'paralog' is a homolog in a different species. Hence, we think that we have used these terms correctly.

As an osmophile, yeasts abundantly occupy ecological niche of leaf surface, fruits and in milk. Do you have any idea which sulfonates may be found in these environments?

Yeasts have been isolated from a great variety of ecological environments, with the same species usually being present in a great diversity of niches. This indicates that yeasts have to be able to adapt to a wide range of nutrient profiles being available in these niches. Hence, not every ecological niche in which a certain yeast species is found is bound to have all nutrients that are relevant for the survival of the species. We do not know what sulfonates may be present on leaf surfaces, in fruits and in milk and in what levels, but these substrates do not necessarily have to contain sulfonates for these compounds to be important for the yeast species. Their abundant presence in soils may already make sulfonate uptake crucial for survival of many if not most yeast species.

Michaelis constant for transport (K_T). Since transporters do not modify substrate as enzymes, it is unprecise to use K_m symbol. Better use Greek tau letter to describe transportation.

K_m values are generally used in the literature and in text books to describe the kinetics of transporters. K_T values are used for the kinetics of substrate assimilation into cellular organic matter. These are two clearly distinct processes. We have only measured transport kinetics using short-term uptake experiments and hence we feel that the use of the term K_m is appropriate in our case.

In text and in figures: may be better use name of loci instead of accession numbers? e.g. *A. nidulans* XP_681611 and CBF80346 refer to one same AN8342 locus.

We have now indicated in the Fig. 4B legend that XP_681611 and CBF80346 are differently predicted splice variants derived from the same locus AN8342.

Line 20 - inorganic (oxidized) sulfur anions rather than inorganic sulfate (sulfite and thiosulfate are not sulfate).

This has been corrected.

Line 167 - figure 3 is not clear to understand which subfigure is related to sulfate uptake.

We have reorganized Fig. 3 and have added panel labels to make the figure more clear. This has indeed greatly improved the understanding of the figure.

Line 188, 534 - Unclear. Is it a gene tree or rather protein tree with protein sequences used?

It is indeed a protein tree. This has been corrected.

Line 235 - *Metarhizium* is a filamentous fungus, not a yeast.

This would have been corrected, but this paragraph has been deleted in response to the request of shortening the text.

Line 237 - AstA (fourth capital letter refer to no locus).

This has been corrected.

Line 286 - butanesulfonate and hydroxymethane sulfonate

This has been corrected.

Line 390 - weak hypothesis. Even under repression, traces of Soa might be lethal in sulfite-containing wine. To prevent sulfite uptake, it would be better for yeast to lose Soa.

We have made this statement more clear: 'Our results suggest that inactivation of *Soa1* by deletion of the *SOA1* gene might offer an additional approach to further enhance sulfite tolerance of wine yeast strains.'

Line 477 - have You checked a negative control how 1.5% MeOH affect on H+ decoupling?

The negative control (0 μ M CCCP) included 1.5% methanol.

We have improved the text:

'The final concentration of methanol was kept constant at 1.5%, also in the negative control.'

Line 506, 547 - means yeast expressing *F. oxysporum* gene, not *Fusarium* itself

We have improved the text:

The cultures of *S. cerevisiae* expressing *F. oxysporum* ENH65823 with L-cysteic acid and 40 mM sulfate as sole sulfur source, displayed an atypical growth behavior.

Table 1 - I assume that should be *S. uvarum* Δ soa1/2 mutants instead of Δ sul1/2

This is correct indeed and we have made the necessary adjustments. We also added the single soa1 Δ strain in the Table 1, which was forgotten previously.

Fig.3 miss slope in taurine and sulfate (SOA) transport plots. In many plots miss standard error bars

We have added the slope of the taurine transport experiment to the curve.

All plots of transport experiments already had error bars for each data point, but in many cases the error is so small that the error bar becomes invisible. The error indicated is from one representative experiment.

We have added a remark in this respect in the Fig.1 and Fig.3 legends:

'In many cases the error is so small that the error bars are not visible.'

Fig.1 and 5 - w/o sulfur added (without slash, it resembles W0, the yeast minimal medium)

We have corrected this error.

References

- Goto-Yamamoto, N., Kitano, K., Shiki, K., Yoshida, Y., Suzuki, T., Iwata, T., . . . Hara, S. (1998). *SSU1-R*, a sulfite resistance gene of wine yeast, is an allele of *SSU1* with a different upstream sequence. *Journal of Fermentation and Bioengineering*, 86(5), 427-433. doi:10.1016/S0922-338x(98)80146-3
- Nardi, T., Corich, V., Giacomini, A., & Blondin, B. (2010). A sulphite-inducible form of the sulphite efflux gene *SSU1* in a *Saccharomyces cerevisiae* wine yeast. *Microbiology*, 156(Pt 6), 1686-1696. doi:10.1099/mic.0.036723-0
- Park, H., & Bakalinsky, A. T. (2000). *SSU1* mediates sulphite efflux in *Saccharomyces cerevisiae*. *Yeast*, 16(10), 881-888. doi:10.1002/1097-0061(200007)16:10<881::AID-YEA576>3.0.CO;2-3
- Pilsyk, S., Natorff, R., Gawinska-Urbanowicz, H., & Kruszewska, J. S. (2015). *Fusarium sambucinum* *astA* gene expressed during potato infection is a functional orthologue of *Aspergillus nidulans* *astA*. *Fungal Biol*, 119(6), 509-517. doi:10.1016/j.funbio.2015.02.002
- Ran, F. A., Hsu, P. D., Wright, J., Agarwala, V., Scott, D. A., & Zhang, F. (2013). Genome engineering using the CRISPR-Cas9 system. *Nat Protoc*, 8(11), 2281-2308. doi:10.1038/nprot.2013.143
- Tweedie, J. W., & Segel, I. H. (1970). Specificity of transport processes for sulfur, selenium, and molybdenum anions by filamentous fungi. *Biochim Biophys Acta*, 196(1), 95-106.
- Wang, K., Ussery, D. W., & Brunak, S. (2009). Analysis and prediction of gene splice sites in four *Aspergillus* genomes. *Fungal Genet Biol*, 46 Suppl 1, S14-18. doi:10.1016/j.fgb.2008.09.010

Wortman, J. R., Gilsenan, J. M., Joardar, V., Deegan, J., Clutterbuck, J., Andersen, M. R., . . . Turner, G. (2009). The 2008 update of the *Aspergillus nidulans* genome annotation: a community effort. *Fungal Genet Biol*, *46 Suppl 1*, S2-13. doi:10.1016/j.fgb.2008.12.003

Reviewer #1 (Remarks to the Author)

The authors have addressed carefully most of the comments of the three reviewers. I have no further comments.

Reviewer #2 (Remarks to the Author)

The authors have adequately addressed my original comments.

Reviewer #3 (Remarks to the Author)

After revision, authors indeed improved the manuscript. Figures and plots are more readable, Km differences are clearly described, additional experimental controls were mentioned. However, some concerns are still worth to correct.

In response authors claim: "The definition of 'ortholog' is a homolog in the same species" - it is wrong. Ortholog refers to the most homologous protein in different species. next: "...while a 'paralog' is a homolog in a different species" - wrong. Paralog is a homolog found in one species (like soa1 and soa2).

Orthologs are genes in different species that evolved from a common ancestral gene by speciation. Normally, orthologs retain the same function in the course of evolution (like *S.cerevisiae* soa1 and *S.arboricola* soa1, not soa2 nor astA). Soa1 orthologs have the same substrate specificity. Homolog refers to related genes separated by the event of speciation or to the genes separated by the duplication (paralogs soa1 and soa2).

Of course it is a minor mistake, however, it leads to misunderstanding that all proteins are orthologous and they transport the same substrate. That's why it is more safely to name them homolog. Please, verify this in literature.

Authors claim that XP_681611 and CBF80346 are differently predicted splice variants derived from the same locus AN8342. It is not true.

Broad Institute resequenced *A. nidulans* genome (after Cereon and Whitehead) and annotated genes as a version 0.2. In this version, intron positions were detected in silico, without any experimental data. Next released versions of genome annotation have numbers 0.3, 0.4, 0.5 and the loci info was corrected by experimental data (RNA-seq and EST sequencing). Currently AspGD database serves as the reference one for genomic data, where the AN8342.5 locus corresponds to old GeneBank Acc. no CBF80346. According to AspGD, in 2010-05-27 The AN8342 gene model was modified by Broad Institute. However, these modifications were rejected from the Version 5 (see: AspGD Sequence Documentation).

Basing on experimental data (RNA-seq, ESTs), alternative splicing of AN8342 is doubtful and the XP_681611 variant, as undetected in nature, is incorrect.

Fig.1 and 5 - w/o sulfur added (still uncorrected slash)

Answers to the comments of the reviewers

Reviewer #1

None

Reviewer #2:

None

Reviewer #3 (Remarks to the Author):

After revision, authors indeed improved the manuscript. Figures and plots are more readable, Km differences are clearly described, additional experimental controls were mentioned. However, some concerns are still worth to correct.

In response authors claim: "The definition of 'ortholog' is a homolog in the same species" - it is wrong. Ortholog refers to the most homologous protein in different species. next: "...while a 'paralog' is a homolog in a different species" - wrong. Paralog is a homolog found in one species (like soa1 and soa2).

Orthologs are genes in different species that evolved from a common ancestral gene by speciation. Normally, orthologs retain the same function in the course of evolution (like *S.cerevisiae* soa1 and *S.arboricola* soa1, not soa2 nor astA). Soa1 orthologs have the same substrate specificity. Homolog refers to related genes separated by the event of speciation or to the genes separated by the duplication (paralogs soa1 and soa2).

Of course it is a minor mistake, however, it leads to misunderstanding that all proteins are orthologous and they transport the same substrate. That's why it is more safely to name them homolog. Please, verify this in literature.

ANSWER. We are sorry for the error in the rebuttal letter, which was an unfortunate oversight. We fully agree with the referee about the definitions of ortholog and paralog. The rebuttal sentence should have read:

'The definition of 'ortholog' is a homolog in a different species, while a 'paralog' is a homolog in the same species.'

In the manuscript, we have used the terms correctly. We have checked this again. The error was only present in the rebuttal sentence.

Authors claim that XP_681611 and CBF80346 are differently predicted splice variants derived from the same locus AN8342. It is not true.

Broad Institute resequenced *A. nidulans* genome (after Cereon and Whitehead) and annotated genes as a version 0.2. In this version, intron positions were detected in silico, without any experimental data. Next released versions of genome annotation have numbers 0.3, 0.4, 0.5 and the loci info was corrected by experimental data (RNA-seq and EST sequencing). Currently AspGD database serves as the reference one for genomic data, where the AN8342.5 locus corresponds to

old GeneBank Acc. no CBF80346. According to AspGD, in 2010-05-27 The AN8342 gene model was modified by Broad Institute. However, these modifications were rejected from the Version 5 (see: AspGDSequence Documentation).

Basing on experimental data (RNA-seq, ESTs), alternative splicing of AN8342 is doubtful and the XP_681611 variant, as undetected in nature, is incorrect.

ANSWER. We agree with the referee and we have improved the text accordingly

The A. nidulans predicted proteins 2 (XP_681611) and 3 (CBF80346) are potential splice variants derived from the same locus AN8342, although the existence of XP_681611 mRNA was rejected from the current gene model (www.aspergillusgenome.org).

Fig.1 and 5 – w/o sulfur added (still uncorrected slash)

ANSWER. This has now been corrected properly both in Fig. 1 and 5.